



# Assessing the potential efficacy of marine cloud brightening for cooling Earth using a simple heuristic model

Robert Wood[1]

[1]Department of Atmospheric Sciences, University of Washington, Seattle, WA 98195, USA

*Correspondence to*: Robert Wood (robwood2@uw.edu)

**Abstract.** A simple heuristic model is described to assess the potential for increasing solar reflection by augmenting the aerosol population below marine low clouds, which nominally leads to increased cloud droplet concentration and albedo. The model estimates the collective impact of many point-source particle sprayers, each of which generates a plume of injected particles that affects clouds over a limited area. A look-up table derived from simulations of an explicit aerosol activation scheme is used to derive cloud droplet concentration as a function of the sub-cloud aerosol size distribution and updraft speed, and a modified version of Twomey's formulation is used to estimate radiative forcing. Plume overlap is accounted for using a Poisson distribution assuming idealized elongated cuboid plumes that have a length driven by aerosol lifetime and wind speed, a width consistent with satellite observations of ship track broadening, and a depth equal to an assumed boundary layer depth. The model is found to perform favorably against estimates of brightening from large eddy simulation studies that explicitly model cloud responses to aerosol injections over a range of conditions. Although the heuristic model does not account for cloud condensate or coverage adjustments to aerosol, in most realistic ambient remote marine conditions these tend to augment the Twomey effect in the large eddy simulations, with the resulting being a modest underprediction of brightening in the heuristic model.

The heuristic model is used to evaluate the potential for global radiative forcing from marine cloud brightening as a function of the quantity, size, and lifetime of salt particles injected per sprayer and the number of sprayers deployed. Radiative forcing is sensitive to both the background aerosol size distribution in the marine boundary layer into which particles are injected, and the assumed updraft speed. Given representative values from the literature, radiative forcing sufficient to offset a doubling of carbon dioxide $\Delta F_{2xCO2}$ is possible but would require spraying over 50% or more of the ocean area. This is likely to require at least $10^4$ sprayers to avoid major losses of particles due to near-sprayer coagulation. The optimal dry diameter of injected particles, for a given salt mass injection rate, is 30-60 nm. A major consequence is that the total salt emission rate (50-70 Tg yr$^{-1}$) required to offset $\Delta F_{2xCO2}$ is a factor of five lower than the emissions rates required to generate significant forcing in previous studies with climate models, which have mostly assumed dry diameters for injected particles in excess of 200 nm. With the lower required emissions, the salt mass loading in the marine boundary layer for $\Delta F_{2xCO2}$ is dominated by natural salt aerosol, with injected particles only contributing ~10%. When using particle sizes optimized for cloud brightening, the aerosol direct radiative forcing is shown to make a minimal contribution to the overall radiative forcing.

## 1 Introduction

Marine low clouds reflect solar radiation and cool Earth as a result (Hartmann and Short, 1980; Ramanathan et al., 1989). The solar radiation reflected by marine low clouds (albedo) increases with the amount of liquid water they contain and as the size of cloud droplets decreases (Stephens 1978). Twomey (1974, 1977) showed that, for a fixed liquid water path (*LWP*), cloud albedo increases with the concentration of cloud droplets ($N_d$). Thus anthropogenic aerosol pollution increases cloud albedo and cools climate. Four decades of subsequent research has established the "Twomey effect" as the largest contributor to the overall cooling impact of aerosols on climate (Zelinka et al., 2014; Bellouin et al., 2020).



In recent decades, evidence has mounted showing cloud macrophysical adjustments to aerosol increases. Albrecht (1989) suggested that reduced droplet sizes would lead to suppressed collision-coalescence, greater retention of water, and an augmentation of the Twomey effect. Modelling and observations do both show precipitation suppression by aerosol in warm clouds (Ackerman et al., 2004; Sorooshian et al., 2010; Terai et al., 2015), and yet observations of ship tracks (Coakley and Walsh 2003, Toll et al., 2019), pollution plumes (Toll et al., 2019; Trofimov et al., 2020), and large-scale shipping lanes (Diamond et al., 2020) reveal LWP reductions in the mean. Modeling has shown that aerosols can cause both positive and negative LWP adjustments (Ackerman et al., 2004; Wood 2007), with the sign of the change dependent on meteorological conditions. Reduced LWP stems from increased cloud top entrainment of dry free-tropospheric air due to smaller cloud droplets and/or turbulent invigoration of the boundary layer caused by suppressed precipitation (Ackerman et al., 2004; Bretherton et al., 2007; Wood 2007). Studies using shipping and land-based pollution sources suggest that mean LWP decreases may offset the Twomey response to a degree that ranges from 3% (Trofimov et al., 2015) to perhaps 20% (Toll et al., 2019; Diamond et al., 2020). LWP adjustments in low clouds are poorly handed in large scale models (Malavelle et al., 2017), which almost universally show LWP increases in simulations of anthropogenic aerosol impacts (Lohmann and Feichter 2005; Isaksen et al., 2009; Bellouin et al., 2020). Global models also tend to show cloud cover increases in response to aerosol, but these appear to be small compared with the Twomey responses and LWP adjustments (Zelinka et al., 2014). Cloud cover adjustments are difficult to constrain using observations (e.g., Gryspeerdt et al., 2016; Possner et al., 2018).

The high sensitivity of cloud albedo to aerosol increases led Latham (1990) to speculate that cloud albedo could potentially be increased deliberately by augmenting the number of aerosol particles ingested into them. This is commonly known as "marine cloud brightening" (MCB), and it has been an increasing focus of research as a potential climate intervention strategy for over a decade (e.g., Latham et al., 2008, 2012; Jones et al., 2009; Rasch et al., 2009; Alterskjær et al., 2012; National Research Council 2015; Ahlm et al., 2017; Stjern et al., 2018). MCB involves spraying small particles of sea salt aerosol into the marine boundary layer (MBL), increasing the concentration of cloud condensation nuclei. This ideally results in a higher concentration of cloud droplets and more reflective clouds. Any large-scale deployment of MCB would involve many point source injections from seagoing vessels distributed over the ocean (Salter et al., 2008). Essentially, such a deployment can be thought of as a deliberate augmentation of the natural experiment currently being conducted by the fleet of commercial ships (~60,000) that are currently emitting aerosol and precursor gases over the world's oceans (Eyring et al., 2010). Thus, we can draw on the study of ship tracks and shipping lanes to provide insights regarding the potential efficacy of MCB.

A ship track is a brightened curvilinear feature in a marine cloud deck caused by the emission of particles and their precursors from an individual ship (Conover 1966). These tracks provide dramatic evidence that cloud reflectivity can increase when particles are released into the MBL. However, ship tracks are insufficient to estimate the large-scale radiative forcing possible. The global increase in reflected shortwave radiation from discernible ship tracks has been estimated from satellite observations





to be ~4-6×10$^{-4}$ W m$^{-2}$ (Schreier et al. 2007), which is 2-3 orders of magnitude smaller than climate model estimates of the
total effect of shipping emissions of aerosol and aerosol precursors, which range from 0.06-0.6 W m$^{-2}$ (Capaldo et al., 1999;
Lauer et al., 2007; Eyring et al., 2010; Peters et al., 2012; Partanen et al., 2013). The most easily discernible ship tracks form
in very shallow MBLs (Durkee et al., 2000). These "Type I" ship tracks tend to occur in MBLs with particularly low
concentrations of background aerosol (Hindman et al., 1994; Ackerman et al., 1995), in which turbulent mixing is weak

because drizzle depletes liquid water and precludes strong cloud top radiative cooling. A more common type of ship track
("Type II") tends to be more readily discernible using near infrared rather than visible satellite imagery (Coakley et al., 1987),
highlighting the smaller droplets in the track. The MBLs in which Type II ship tracks form tend to be somewhat deeper, more
well-mixed and strongly driven by cloud top cooling. Ship track albedo perturbations in these cases tend to be weaker than in
Type I tracks. Large eddy simulations of deep stratocumulus-topped MBLs indicate that albedo can be increased substantially

by injected aerosol emissions even when a clear track is not discernible (Possner et al., 2018). In Durkee et al. (2000) no ship
tracks were detected in MBLs deeper than 800 m, but Possner et al. (2020) show that over 80% of all stratocumulus-topped
MBLs over the oceans are deeper than 800 m, where surface emissions can increase cloud albedo but tracks may not be easy
to detect.

An alternative to observational studies of individual ship tracks is to quantify the mean radiative forcing over a heavily
trafficked area to assess the aggregate effect of shipping. Diamond et al. (2020) was able to discern a corridor of enhanced
mean $N_d$ in clouds above a shipping lane that traverses the SE Atlantic subtropical stratocumulus deck. In this corridor, an
increase in reflected diurnal-seasonal mean shortwave radiation of 2 W m$^{-2}$ was observed associated with an increase in $N_d$
of ~5 cm$^{-3}$, which is consistent with expectations from the Twomey effect. Cloud adjustments were found to be relatively

small, with reduced cloud LWP in the shipping lane offsetting ~20% of the Twomey effect, and a small cloud fraction increase
augmenting the Twomey effect by ~10%. Although the radiative forcing would need to be somewhat stronger for MCB to
offset a significant fraction of the radiative forcing from increased greenhouse gases, the lack of major cancelling cloud
adjustments points to the potential for regional albedo enhancement using MCB. In this case the aerosols (from ship emissions)
were inadvertently brightening clouds; aerosols of a size and concentration that targets intentional cloud brightening would

very likely have a larger impact on cloud albedo and radiative forcing.

Climate models demonstrate the potential for producing a globally-significant radiative forcing from MCB. These studies fall
into two broad categories: (i) studies in which $N_d$ (or droplet effective radius $r_e$) in some fraction of the marine low cloud
population is altered to some specified value in order to increase cloud albedo; (ii) studies that achieve cloud albedo changes

by increasing the surface aerosol source and treating the aerosol activation process, leading to changes in $N_d$. The latter studies
involve a more complete treatment of the chain of causality that links aerosol emissions to brightening, while the former studies
can be carried out without explicit representation of aerosol-cloud interaction processes.




Seeded regions in studies with specified $N_d$ or $r_e$ perturbations have increased $N_d$ to different levels: 375 cm$^{-3}$ in Jones et al.,
(2009), 1000 cm$^{-3}$ in Rasch et al., (2009) and Baughman et al., (2012) and both 375 and 1000 cm$^{-3}$ in Latham et al. (2008). In
Bala et al. (2011), the cloud effective radius is instead decreased from 14 to 11.5 μm for all marine liquid clouds, which is
approximately equivalent to increasing $N_d$ by 80%. Stjern et al. (2018) increase $N_d$ by 50% in all marine low clouds. Because
cloud albedo increases scale with the ratio of perturbed (seeded) to unperturbed $N_d$ (Sect. 2.1), these changes represent a wide
diversity in how much a seeded cloud is brightened in each study. Very different fractions of the available ocean are seeded in
different studies, ranging from 1.0, 1.6, 2.1 and 4.7% of the ocean area in Jones et al., (2009), 9% in Baughman et al. (2012),
20, 30, 40 and 70% in Rasch et al., (2009) and the entire ocean in Bala et al. (2011). Jones et al. (2009) achieved a forcing of
-1 W m$^{-2}$ despite only perturbing 4.7% of the ocean surface, but perturbed regions had extensive low clouds. Rasch et al. (2009)
went further and identified the albedo susceptibility (change in albedo upon increasing $N_d$ to 375 cm$^{-3}$) for each grid box on a
seasonal basis. The most susceptible 20, 30, 40 and 70% of the boxes were then used as seeding regions. The wide range in
different areas seeded and in the strength of the $N_d$ perturbation where the seeding occurs makes it difficult to intercompare
the effectiveness of the seeding across studies.

Climate model studies in which an aerosol surface source is added as a proxy for deliberate spraying have also been shown to
produce globally significant radiative forcing (Ahlm et al., 2017), with values in some studies more than offsetting that from
doubling $CO_2$ (e.g., Alterskjaer et al., 2012). Such studies introduce a number of additional degrees of freedom into the
experimental design. A comprehensive representation of the aerosol lifecycle is needed, as well as an aerosol activation
parameterization to predict $N_d$ as a function of the aerosol size distribution in the MBL. As studies with aerosol activation
schemes and/or parcel models have shown, $N_d$ is sensitive primarily to the concentration of aerosol in the accumulation mode
(particles with dry diameters around 50-200 nm), but is also sensitive to updraft speed and to small concentrations of coarse
mode aerosol, which reduce the peak supersaturation in an updraft and lower the fraction of smaller aerosols activated (Ghan
et al., 1998; McFiggans et al., 2006).

An additional aerosol surface source from an MCB sprayer can, in principle, be tailored to consist of particles of a specific
diameter. Connolly et al. (2014) explored the optimal particle size given energy constraints on particle production, which
primarily scales with the mass of salt injected, and found that sodium chloride particles with a modal diameter in the range 30-
90 nm are optimal. Climate model studies to date have typically introduced injected particles with modal diameters that are
several times as large as this (Altersjkaer et al., 2012; Ahlm et al., 2017), which implies that these models likely require much
larger salt mass emissions than may be required if smaller particles are injected. Only Partanen et al. (2012) has tested the
sensitivity to injecting particles with modal dry diameter of 100 nm and found the same brightening as in a base case with 200
140  nm diameter particles but with ~5 times less mass injected. Consideration of total salt mass injected is important not only from
the perspective of the energy required to produce particles, but also because major increases in sodium chloride aerosol mass
could potentially alter natural chemical cycles in the MBL (Horowitz et al., 2020).





This study describes a simple heuristic model that predicts the global radiative forcing from MCB using physical principles to determine the collective impact of plumes from many point source sprayers distributed over the oceans on $N_d$ and cloud albedo. The model is designed to facilitate easy experimentation on the factors controlling MCB, including details of the unperturbed aerosol size distribution, the number concentration, size and residence time of injected particles, the number of sprayers, and the fraction of the ocean over which sprayers are deployed. Section 2 describes the heuristic model in detail, and Sect. 3 tests the model using comparisons with high resolution, small-domain large eddy simulation models into which point source injections are introduced. Section 4 uses the heuristic model to examine factors controlling global radiative forcing from MCB, and critically examines some assumptions made in previous climate model studies. Finally, Sect. 5 discusses implications of the results and suggests pathways for future study, and Sect. 6 provides conclusions.

## 2 Heuristic model description

### 2.1 Radiative forcing from aerosol-cloud interactions

Central to the model is Twomey's formulation for the susceptibility of cloud albedo $\alpha_c$ to an increase in $N_d$ assuming no cloud adjustments (Twomey 1977), viz.

$$\frac{d\alpha_c}{dN_d} = \frac{\alpha_c(1-\alpha_c)}{3N_d} \tag{1}$$

Integrating Eq. (1) gives an expression for the increase in cloud albedo $\Delta\alpha_c$ caused by an increase in $N_d$:

$$\Delta\alpha_c = \frac{\alpha_c(1-\alpha_c)\left(r_N^{1/3}-1\right)}{1+\alpha_c\left(r_N^{1/3}-1\right)} \tag{2}$$

where $r_N = N_d'/N_d$ is the ratio of the droplet concentration in seeded vs unperturbed clouds. It is worth noting that Eq. (2) is rather insensitive to $\alpha_c$, such that $\Delta\alpha_c$ varies by only ~10% as $\alpha_c$ changes from 0.3-0.7. Thus, the key sensitivity in Eq. (2) is to the value of $r_N$.

To estimate the top-of-atmosphere (TOA) albedo for the same cloud requires a conversion to account for the absorption and scattering of solar radiation by the atmosphere above cloud. We follow the approach by Diamond et al. (2020, Eq. 17) and multiply the cloud albedo change by an atmospheric correction factor $f_{atm}$:

$$\phi_{atm} = \frac{\Delta\alpha_{c,TOA}}{\Delta\alpha_c} = \frac{T_{FT}^2}{(1-\alpha_{FT}\alpha_c)^2} \tag{3}$$

Where $T_{FT}$ and $\alpha_{FT}$ are the transmissivity and albedo of the free troposphere only. The more variable of these two parameters is $T_{FT}$, which depends upon free-tropospheric water vapor. Here, we assume a value of $T_{FT} = 0.8$, consistent with values over dry regions of the Tropics and midlatitudes from the CERES-SYN product (Doelling et al., 2013). Free-tropospheric albedo is less variable and we here assume a value of $\alpha_{FT} = 0.06$ (also consistent with CERES-SYN). For typical cloud albedos $\alpha_c$ in



the range 0.25 to 0.75, $\phi_{atm}$ ranges from 0.66 to 0.70, so for simplicity we herein assume $\phi_{atm} = 0.70$. We estimate TOA indirect radiative forcing as $-F_\odot \phi_{atm} \Delta\alpha_c$, where $F_\odot$ is the mean incoming solar irradiance averaged over day and night. Here,

we assume a value of $F_\odot$ equal to the global mean solar irradiance $F_\odot = 342$ W m$^{-2}$. Geographical variation in insolation is not considered.

## 2.2 Regions where sprayers operate

Marine cloud brightening, by definition, would only be deployed over the fraction of Earth covered by ocean. We further restrict this area to minimize the likelihood that plumes will intersect land areas by limiting spraying to $10 \times 10°$ boxes that

contain less than 10% land area. The choice of boxes 10° on a side is made because plumes are of order 1000 km in length (see Sect. 2.3). This limits the eligible fraction of Earth's surface for spraying, $f_{ocean}$ to 0.54. We then assume that sprayers are confined to operate within some fraction $f_{spray}$ ($0 < f_{spray} \leq 1$) of this eligible area. When $f_{spray} < 1$, we make the assumption that sprayed regions will be those with the highest climatological low cloud cover. To determine the mean low cloud cover for the sprayed subregions $f_{low}$, climatological monthly mean low cloud fractions are determined using MODIS Terra and Aqua

Level 3 liquid cloud fractions (years 2006-2010) for $10 \times 10°$ boxes. As $f_{spray}$ decreases, the regions sprayed consist of those regions with a greater coverage of low clouds. When $f_{spray}$ is very small, spraying occurs only in regions with the highest climatological monthly mean cloud cover. The MODIS data are well-fitted with the empirically-determined expression:

$$f_{low} = 0.32 + 0.36 \exp\left(-3.2 f_{spray}^{0.75}\right) \qquad (4)$$

## 2.3 Expression for global radiative forcing associated with MCB

In the sprayed areas without low clouds, MCB exerts no radiative forcing from aerosol-cloud interactions. Cloud condensate and coverage adjustments to injected aerosol are assumed to be zero, so that MCB indirect radiative forcing arises only from the Twomey effect. The direct radiative forcing from injected aerosol in cloud-free regions between clouds is quantitatively estimated (see Sect. 2.7), but increasing direct radiative forcing is not a goal of the injection design. The global mean shortwave

radiative forcing $\Delta F$ from MCB aerosol-cloud interactions is written as

$$\Delta F = -F_\odot f_{ocean} f_{spray} f_{low} \phi_{atm} \Delta\alpha_c \qquad (5)$$

To give a "back of the envelope" assessment of the potential for MCB, we take $f_{ocean} = 0.54$, assume $f_{spray} = 1$ and use Eq. (4)

to set $f_{low} = 0.33$. If cloud albedo is increased by $\Delta\alpha_c = 0.01$, then $\Delta F = -0.41$ W m$^{-2}$. Alternatively, it would take a cloud albedo increase of $\Delta\alpha_c = 0.09$ to produce a radiative forcing of $-3.7$ W m$^{-2}$, which would balance the longwave radiative forcing $\Delta F_{2\times CO2}$ from doubling CO$_2$. Figure 1 shows $\Delta F$ as a function of $r_N$ for different values of $f_{spray}$ and $\alpha_c$. Using Eq. (2), if we assume $\alpha_c = 0.56$ (Bender et al. 2011 finds TOA cloud albedos of 0.35 to 0.42 for overcast stratocumulus in the major


subtropical Sc decks, which must be corrected to cloud albedos with Eq. 3), then the ratio of seeded to unseeded cloud droplet
concentration ($r_N = N_d'/N_d$) would need to be 3.0 to produce a forcing with a magnitude equal to $\Delta F_{2 \times CO2}$. Assuming the entire
ocean area could be seeded, we find a value of $r_N$=2.4, which is in the range of $N_d$ increases over the ocean (2.10-2.85) that
were needed to counter $CO_2$ doubling in an analysis of three variants on a climate model (Slingo 1990). If only half of the
eligible ocean area is seeded (i.e., $f_{spray} = 0.5$), then $r_N$ would need to be at least 7 to counter $CO_2$ doubling (Fig. 1). Stjern et
al. (2018) analyzed an ensemble of different climate models in which $N_d$ for all marine low clouds is increased by 50% ($r_N$=1.5)
as a proxy for MCB, and found an ensemble mean $\Delta F$ = -1.9 W m$^{-2}$. Based on Fig. 1, and scaling the forcing to include the
entire ocean, Eq. (5) produces a very similar forcing $\Delta F$ = -1.8 W m$^{-2}$. This is also consistent with the models in Stjern et al.
(2018) having small cloud adjustments overall, so that the overwhelming bulk of the forcing is from the Twomey effect.

**2.4 Aerosol delivery and plume/track configuration**

Any practical MCB deployment would be unable to produce uniform increases of $N_d$ because seeding is necessarily discrete
in nature, rather than distributed evenly. It is impractical to deploy sprayers at every point over the ocean; in practice, any
deployment would likely consist of an array of floating particle injection systems distributed throughout regions where low
clouds occur. To extend the heuristic model to account for this, assumptions are made about the spatiotemporal extent of the
region affected by a single sprayer. Sprayers are assumed to be stationary so that air masses pass over them at the rate of the
near-surface wind speed $U_0$, which is taken as 7 m s$^{-1}$, the mean value over oceans (Archer and Jacobson 2005). Each sprayer
injects sodium chloride particles continuously with a salt mass rate $\dot{M}_s$. Injected particles have a lognormal size distribution
with geometric mean dry diameter (GMD) $D_s$ and geometric standard deviation (GMS) $S$. The total number of particles
sprayed per second from each sprayer $\dot{N}_s$ is then

$$\dot{N}_s = \frac{6 \dot{M}_s}{\pi \rho_s D_s^3 e^{9(\ln S)^2/2}} \tag{6}$$

where $\rho_s$ is the density of solid sodium chloride (2160 kg m$^{-3}$). The volume into which particles are emitted increases with
time as the plume expands to fill the depth of the MBL and widens horizontally. The timescale for vertical dispersion through
the depth of the MBL is 10-20 min (Chosson et al., 2008), as evidenced by the fact that, in ship tracks, brightened clouds
become evident typically 10-20 km downwind of the responsible ship. As satellite data readily show, ship tracks from
commercial shipping are narrower close to the emitting ship and broaden downstream (Durkee et al., 2000). After rapid vertical
dispersion through the MBL, dilution primarily occurs through lateral diffusion. Entrainment of lower concentration free-
tropospheric air also dilutes the plume, but at a slower rate. The lateral track broadening rate is highly variable but is
parameterized using the Heffter (1965) broadening rate $K$=1.85 km hr$^{-1}$ (see Fig. 7 in Durkee et al. 2000). This rate is broadly
consistent with large eddy simulations of horizontal tracer spread in the cloudy MBL (Wang et al., 2011).

Injected particles in the model have a characteristic residence e-folding timescale $\tau_{res}$. This residence time incorporates a
number of processes influencing particle lifetime, including removal by coalescence scavenging, scavenging by clouds and





aerosol particles and dry deposition. The value of $\tau_{\text{res}}$ varies with meteorological conditions, cloud and precipitation properties, and is also expected to be somewhat size dependent. In regions of marine stratocumulus regions values of $\tau_{\text{res}}$ of 2-3 days are consistent with estimates of precipitation scavenging (Wood et al., 2012), and $\tau_{\text{res}} = 2$ days is used as standard.

After a time $t$, the particles injected at time $t = 0$ have moved a distance $x = U_0 t$. Taking into account both dilution and removal processes, given a plume width $W(t) = Kt$ and assuming dispersion through the entire MBL depth $h$, the injected particle concentration $N_s(t)$ at time $t$ is:

$$N_s(t) = \frac{\dot{N}_s}{U_0 h K t} e^{-t/\tau_{\text{res}}} \qquad (7)$$

The wind speed and residence time define a length scale ($L_t = U_0 \tau_{\text{res}}$) that effectively determines the streamwise length scale over which the particle concentration is affected by spraying. The area over the Earth's surface perturbed by each sprayer $A$ is then determined by multiplying this length scale by a characteristic track width $W_t$, i.e. $A_t = L_t W_t = U_0 \tau_{\text{res}} W_t$. A linearly widening plume/track will expand to a width $K\tau_{\text{res}}$ over the lifetime of the particles.

To estimate radiative forcing, the injected particle concentration $N_s(t)$ is added to an assumed background aerosol over the entire track area and over the depth $h$ of the MBL. Aerosol activation to form cloud droplets is carried out for the background aerosol and for the perturbed (background+injected) aerosol using an assumed updraft speed. Section 2.6 provides details of the activation scheme, and the aerosol physical and chemical properties. The ratio of the perturbed to background cloud $N_d$ from the activation scheme are used in the calculation of radiative forcing (Eq. 2 and 5).


Figure 2 shows results from the model for a laterally-spreading track, along with injected particle concentration and additional reflected shortwave from cloud brightening as a function of time/distance downstream of a point-source sprayer. The heuristic model assumes an elongated cuboid plume (fixed width, height and length) with the plume length $L_t = U_0 \tau_{\text{res}}$ and plume width taken to be the width of the linearly expanding plume at time $\tau_{\text{res}}/2$, i.e., $W_t = K\tau_{\text{res}}/2$. The (time-independent) number

concentration $N_{s1}$ of injected aerosol particles in the cuboid plume is

$$N_{s1} = \frac{\dot{N}_s}{h U_0 W_t} = \frac{2\dot{N}_s}{h U_0 K \tau_{\text{res}}} \qquad (8)$$

It is relatively straightforward to show that the overall injected particle concentration integrated over time is the same for the laterally-spreading track (Eq. 7) and the cuboid track (Eq. 8). Although the reflected solar energy from the two tracks is not

identical (Fig. 2c), the values are found to be close. A heuristic model track reflects slightly less than a spreading track for a given spray rate, with the difference growing as the magnitude of the $N_d$ perturbation increases.



Experimentation with different spray, background aerosol and cloud configurations shows that the reflected sunlight for the cuboid track is within 5% of that for the spreading track for number spray rates $\dot{N}_s < 10^{16}$ s$^{-1}$, with the cuboid model track

being slightly less reflective. As $\dot{N}_s$ increases, the ratio of the additional energy reflected by the spreading track to that from the cuboid track increases steadily, reaching 1.2 for $\dot{N}_s = 5\times10^{16}$ s$^{-1}$, and 1.5 for $\dot{N}_s = 10^{17}$ s$^{-1}$ with the exact value dependent upon the background aerosol. As the magnitude of the aerosol number perturbation increases, an increasing fraction of the energy reflected occurs at times $t > \tau_{res}$ in the spreading plume. The albedo response in the cuboid track is relatively saturated due to the high aerosol/droplet concentrations (see Fig. 1), so the dilute but widespread aerosol in the spreading plume later on

is more efficient at brightening. As we show in the discussion, coagulation losses during the high concentrations near to the sprayer are likely to be large for particle spray rates much greater than ~$10^{16}$ s$^{-1}$. The cuboid tracks are thus a sufficiently accurate representation of reality to use them in the heuristic model and simplifies the treatment of overlapping tracks.

**2.5 Overlapping tracks**

Given the plume dimensions for the heuristic model tracks, we estimate that the number of (non-overlapping) tracks required

to cover the 54% of the ocean eligible for spraying (~$2.8\times10^{14}$ m$^2$), assuming $W_t$= 44 km, $\tau_{res} = 2$ days, and $U_0 = 7$ m s$^{-1}$ (i.e., $L_t \approx$1200 km; $A_t = 5.28\times10^{10}$ m$^2$) is 5300. If this number of sprayers was to be deployed either randomly or uniformly then overlapping tracks would be unavoidable because air mass trajectories are not constant in time. Monte Carlo simulations were conducted, placing $N_t$ randomly-oriented or aligned rectangular tracks at random over a large domain of area $A$. The probability $p(n)$ of $n$ tracks overlapping in the domain is well-predicted by a Poisson distribution:


$$p_P(n) \approx \frac{\zeta^n e^{-\zeta}}{n!} \qquad (9)$$

where $\zeta = N_t A_t / A$ is the mean track density, i.e., the mean number of superimposed tracks (Fig. 3). Although not shown, $p(n)$ is insensitive to both the track aspect ratio ($L_t/W_t$) and whether the tracks are aligned with their long sides in one direction or

are randomly oriented.

The use of the Poisson distribution makes it straightforward to account for track overlap in the heuristic model; the injected particle concentration $N_s$ at any given location is a multiple $n$ ($n \geq 0$) of the single-track value $N_{s1}$ from Eq. (8):


$$N_s = nN_{s1} = \frac{2\dot{N}_s n}{hU_0 K\tau_{res}} \qquad (10)$$

where the probability of $n$ is given by Eq. (9). For low mean track densities $\zeta$, the most likely value of $n$ is zero (Fig. 3), and the ratio of the standard deviation to the mean value of $n$ is high. A Poisson distribution has equal mean and variance, so the





relative spatial heterogeneity of $N_\text{s}$, i.e., the ratio of the standard deviation to the mean track density, decreases as $\zeta^{-1/2}$.
Because of the concave relationship between $\Delta\alpha_c$ and $N_\text{s}$ (see e.g., Carslaw et al., 2013), a more homogeneous distribution of $N_\text{s}$ over the seeded area will yield a radiative forcing with a larger magnitude for the same mean value of $N_\text{s}$.

**2.6 Aerosol activation and physical and chemical properties**

Aerosol activation to form cloud droplets is treated using a five-dimensional look-up table derived from over 6000 numerical Lagrangian parcel model simulations (see Appendix). In comparing with those using the Abdul-Razzak and Ghan (2000)
quasi-analytical activation scheme (henceforth ARG), we find significant differences that indicate a major underprediction of $N_\text{d}$ with the ARG scheme when injected dry particle diameters are smaller than ~200 nm (see Appendix and also Sect. 4.2), and so use the look-up table to treat activation in the heuristic model.

The aerosol size distributions used are exactly the same for each activation approach. The background (unperturbed) aerosol
particles are assumed to comprise lognormal accumulation and coarse modes. Acccumulation mode size values ($D_{0,\text{acc}}$=175 nm; $S_{0,\text{acc}}$=1.5) are taken from the synthesis of marine accumulation mode measurements by Heintzenberg (2000). Measured marine accumulation mode number concentrations $N_{0,\text{acc}}$ vary considerably over the ocean, and impacts of this on brightening are explored in Sect. 4.2. Although there is significant variability in the composition of marine CCN, studies tend to find that the accumulation mode aerosol in the unpolluted MBL consists of a mixture of sulfate, sea salt, and organic species. Different
assessments of the hygroscopicity parameter ($\kappa$, from Petters and Kreidenweis 2007) of CCN in the marine PBL provide a significant diversity of values, from values as low as 0.45 (Wex et al., 2010) to ~0.7 (Andreae and Rosenfeld 2008). Here, we use the mean marine value of 0.7 from the model study of Pringle et al. (2010) for the unperturbed accumulation mode. The background coarse mode is lognormal with GMD $D_{0,\text{coarse}}$=615 nm and GMS $S_{0,\text{coarse}}$=1.8 taken from summertime measurements at Graciosa Island in the Azores (Zheng et al. 2018). The presence of the coarse mode suppresses the peak
supersaturation in the updraft, increasing the minimum size of the particles that are activated, reducing the activated fraction (Ghan et al. 1997). This is explored further in Sect. 4.2. Injected aerosols are sodium chloride ($\kappa = 1.2$, Petters and Kreidenweis 2007), distributed lognormally with GMD $D_\text{s}$ and GMS $S$, where $D_\text{s}$ is allowed to vary, and $S$=1.6. Table 1 provides a summary of the assumed aerosol properties used.

For most of the analysis presented in this study, a fixed updraft speed of $w$ =0.4 m s$^{-1}$ is assumed in the activation scheme. This is broadly representative of updrafts in the stratocumulus-topped MBL (Nicholls and Leighton 1986; Wood 2005; Bretherton et al., 2010; Zheng et al., 2016). Sensitivity to updraft speed is explored in Sect. 4.3. For simplicity, the temperature and pressure are set to be 280 K and 925 hPa respectively, but the results are not highly sensitive to these values.





### 2.7 Aerosol direct radiative forcing

The heuristic model is also used to produce rough estimates of the aerosol direct radiative forcing from the injected aerosol. We assume direct forcing only in clear sky regions. In an analogous formulation to Eqn. 5, we estimate the global mean direct radiative forcing as:

$$\Delta F_{\mathrm{direct}} = -F_{\odot} f_{\mathrm{ocean}} f_{\mathrm{spray}} f_{\mathrm{clear}} E \tau_{\mathrm{spray}} \tag{11}$$


where $f_{\mathrm{clear}}$ is the clear sky fraction in the regions where sprayers operate, $\tau_{\mathrm{spray}}$ is the aerosol optical thickness (550 nm) of injected particles, and $E$ is the clear-sky radiative forcing efficiency. We use $E$ = -29 W m$^{-2}$ $\tau^{-1}$, which is the average over oceans for a number of models in the AeroCom study of Schulz et al. (2006). In this study, direct effects are only estimated for the case where $f_{\mathrm{spray}}$=1, i.e., sprayers operate in all eligible regions of the oceans, and so $f_{\mathrm{clear}}$ = 0.32 is taken as the

complement of the total cloud cover during the daytime over the global oceans from Hahn and Warren (2007). To estimate $\tau_{\mathrm{spray}}$, the injected aerosol lognormal size distribution (accounting for overlapping tracks as discussed in Sect. 2.5) is used to estimate extinction $\sigma_{\mathrm{spray}}$ at 550 nm using the Mie code of Bohren and Huffman (1998). A mean relative humidity in clear sky MBLs of 80% is used to set a hygroscopic diameter growth factor for sodium chloride of 2.0 from Tang (1996). The assumed MBL depth $h$ = 1 km (Table 1) is used to determine $\tau_{\mathrm{spray}} = h\sigma_{\mathrm{spray}}$. Direct forcing estimates are presented in Sect. 4.4.

### 3. Comparison of heuristic model with large eddy simulations


A number of existing studies in the literature have used large eddy simulations (LES) to explore impacts of salt aerosol injections on marine low cloud microphysical and macrophysical properties and albedo. In contrast to climate models, LES explicitly resolves the turbulent dynamics responsible for aerosol distribution through the PBL including ingestion into clouds, in addition to determining cloud macrophysical responses to aerosol resulting from changes in precipitation and mixing with

the free troposphere. Although it is not currently possible to run LES with domain sizes large enough to examine regional and global MCB, their faithful representation of injections into domains on scales of a few tens to a few hundred km can provide important insights into the potential efficacy of MCB.

The heuristic model framework is adapted to account for the limited LES domain size in order to test its predictions. This also

allows a quantitative intercomparison of the LES results, which is needed because there is a considerable diversity in the domain sizes, spray rates and particle sizes, as well as in the unperturbed cloud states, boundary layer depths, simulation durations, and in the way in which injections are introduced into the domains, across the different LES studies to date (see Table 2). Each LES experiment consisted of an unperturbed (control) case with no particle injections, and a case with particle injections. A total of 18 different injection experiments are extracted from five studies.






Radiative forcing driven by particle injections is estimated for the LES case studies using albedo changes given in the various papers. Unless otherwise stated, heuristic model parameters are those in Table 1. Diurnal mean insolation is assumed. Where appropriate, cloud albedo changes are corrected to the TOA using a fixed value of $\phi_{atm}$ (Eq. 3, Table 1) consistent with that used in the heuristic model. The heuristic model uses a fixed value for unperturbed cloud albedo (Table 1), but the PBL depth

is set to the value for each of the LES cases (Table 2). In several of the cases, the sprayer passes through the model domain multiple times, and in other cases the track does not extend over the entire domain. This is handled in the heuristic model as described in Sect. 2.5. Wang et al., (2011, henceforth W11) also included a simulation where the same rate is injected uniformly over the model domain, as a comparison experiment against a point-source sprayer. This is represented in the heuristic model by assuming a large number of (weaker) sprayers operating in the domain.


Unperturbed (control case) aerosol size distributions for the heuristic model comprise an accumulation mode with distribution parameters from Table 1 (which are close to those assumed in the LES studies) and concentrations are adjusted to produce unperturbed $N_d$ values reported in the various studies with a fixed 0.4 m s$^{-1}$ updraft (i.e., the standard value used in the heuristic model). Jenkins et al. (2013, henceforth J13), used a bin aerosol scheme rather than a modal scheme. Including an additional

coarse mode with modal diameter 500 nm, GSD of 1.8 and concentration of 10 cm$^{-3}$, which we found represents a fairly good match to the size distributions shown in J13, made less than a 10% difference in the radiative forcing predicted by the heuristic model. No coarse mode is included in the heuristic model for the other cases, because none was included in the LES simulations.

Figure 4 presents results from the comparison of the LES and the heuristic model. Overall, the radiative forcing in the LES correlates quite well ($r = 0.62$) with predictions from the heuristic model (Fig. 4a), but the heuristic model underestimates the magnitude of the LES forcing by ~30% in the median. This underestimation is greatest when the unperturbed value of $N_d$ is low (Fig. 4b). There is little bias for cases with $N_d$ ~50 cm$^{-3}$, but there is underprediction of ~2 for $N_d$~10 cm$^{-3}$, and an overprediction of brightening for high $N_d$ cases. The sensitivity of the heuristic model brightening bias to unperturbed

$N_d$ (Fig. 4b) is not driven by a model failure to represent the Twomey effect, as the heuristic model ability to predict domain-mean perturbed $N_d$ is excellent ($r = 0.91$, Fig 4c), with only a 25% overestimate in the median. This would lead to a small (~10%) overprediction of Twomey forcing magnitude. Instead, the heuristic model underprediction at low $N_d$ occurs because particle injection into very clean MBLs often leads to increases in liquid water path (LWP), cloud cover, or both. In these cases, the Twomey effect is augmented by cloud adjustments that result in stronger brightening, and this is not represented in

the heuristic model. The reasons for overprediction of brightening for high $N_d$ cases is unknown, and warrants further attention using more LES studies.



Forcing normalized by the total number of particles injected helps account for the different quantities of particles injected in different studies and provides a useful metric of brightening obtained per particle injected. The LES results show a remarkably

strong dependence of this on the unperturbed $N_d$ (Fig. 4d), with a factor of 20 less brightening as unperturbed $N_d$ increases from 10 to 100 cm$^{-3}$. Although the heuristic model underpredicts (overpredicts) brightening in the clean (polluted) cases, there is still a strong decrease (~factor 10) in the brightening as unperturbed $N_d$ increases (Fig. 4e), as anticipated from the Twomey formulation (Eq. 2). In Figs. 4d and 4e, J13 stands out as an anomaly, with a much weaker per-particle brightening compared with the other models. This appears to be because the injection rates used were greater than is needed. An experiment with the

precipitating case with an injection rate reduced by a factor of 5 (cyan triangles in Figs. 4d and 4e) leads to less than a 15% reduction in brightening, implying asymptotic brightening as injection rates are increased, and little benefit from the high spray rates used in most of the cases in J13.

It is instructive to compare the brightening obtained per mass of salt injected, and Fig. 4f highlights just how wide a spread

there is in this quantity. The most "efficient" brightening is obtained in the Chun et al. (2021) cases with the smallest injected particles (Fig. 4f). For reasons discussed in the introduction, if a forcing can be achieved by injecting less salt mass, then this is desirable, so understanding the optimal size and concentration of injected particles to achieve a required forcing should be a focus for LES studies. These issues are explored further for the heuristic model in Sect. 4.

To synthesize the findings reported here, it should be noted that all the LES studies surveyed show some level of brightening when aerosol injections are introduced. The brightening achieved in the LES experiments, which is here expressed as an equivalent diurnal mean, ranged from ~1-100 W m$^{-2}$, with mean of 17 W m$^{-2}$ and a median of 7 W m$^{-2}$. The median unperturbed cloud $N_d$ (29 cm$^{-3}$) across all the cases here, is somewhat lower than satellite estimates of average values for low clouds over the global oceans (40-90 cm$^{-3}$, Bennartz 2007). We also used the approach of Bennartz (2007) to derive a pdf of estimated $N_d$

for all marine low clouds from MODIS data and found a median value of 50 cm$^{-3}$. Thus, we might anticipate that the clouds simulated in the LES cases have a somewhat greater susceptibility to brightening than the "average" marine cloud. The brightening in the deepest MBL case here (P11) does not stand out as being anomalously weak compared with similarly clean cases, although no clear track is produced in the simulated cloud field (Possner et al., 2018). It is important to stress, however, that several low cloud systems (e.g., self-aggregated cumulus, midlatitude stratus) that contribute significantly to low cloud

cover over the global oceans are not represented in the LES cases in the literature to date. The LES cases also do not provide sufficient constraints on how brightening changes with injection rate and injected particle size across the different meteorological conditions. Another consideration is that almost all of the studies here examine responses that take place within the first day after injection. As Fig. 2 suggests, a significant fraction of the reflected energy likely takes place between 1-3 days after injection. However, studies suggest that cloud adjustments to aerosol may change significantly over timescales of

hours to a few days (Wood 2007; Gryspeerdt et al., 2019; Glassmeier et al., 2021), resulting in changes on longer timescales that may, in some cases, offset some of the Twomey brightening. Thus, although the LES simulations here provide some





validation of the heuristic model, there is a need for many more simulations to test its sensitivities under the full range of meteorological, background aerosol, and aerosol injection scenarios.

## 4. Global forcing estimates from the heuristic model

The heuristic model is next used estimate the global radiative forcing for MCB under different assumptions regarding the number of sprayers and the rate and size of the injected particles. Sensitivity of the forcing to the properties of the background aerosol and updraft speed is also explored.

### 4.1 Sprayer number and injection rate

Figure 5 presents results as a function of the number of sprayers $N_{\mathrm{sprayers}}$ and the salt mass injection rate per sprayer $\dot{M}_{\mathrm{s}}$. For

this case, injected particles have $D_{\mathrm{s}}$=100 nm, and spraying occurs over all eligible ocean areas ($f_{\mathrm{ocean}}$=0.54 and $f_{\mathrm{spray}}$=1 in Eq. (5)). A background accumulation mode aerosol concentration $N_{0,\mathrm{acc}}$=100 cm$^{-3}$ is assumed, representative of conditions over the open oceans (Heintzenberg et al., 2000), and a representative coarse mode is included (Sect. 2.6 and Table 1). Other parameters are set to the values provided in Table 1 and discussed in Sect. 2. The assumed PBL depth of 1000 m is representative of typical conditions in which stratiform marine low clouds occur. A global mean radiative forcing magnitude

of 1-4 W m$^{-2}$ can be achieved, with forcing generally increasing as total salt mass injection rates increase from ~10 to ~60 Tg yr$^{-1}$ (Fig. 5a). As the total injection rate increases beyond 100 Tg yr$^{-1}$ there are somewhat diminishing returns in terms of further brightening, and $-\Delta F$ reaches ~5-8 W m$^{-2}$ for an injection rate of 1000 Tg yr$^{-1}$. The reduced sensitivity as more particles are injected is driven by increased competition for water vapor in the updraft, resulting in a decreasing fraction of injected aerosols activated to form droplets (Fig. 5b, dotted contours). When the injected aerosol concentration is less than

a few hundred cm$^{-3}$, such competition for vapor is relatively modest, and the activated fraction exceeds 70%, but this reduces to only 30-40% at injection rates of 300 Tg yr$^{-1}$.

Given that $-\Delta F$ increases approximately as a function of total mass injection rate for mass injection rates of <50 Tg yr$^{-1}$ (Fig. 5a), roughly the same forcing can be achieved either with a smaller number of high throughput sprayers, or a larger

number of somewhat weaker sprayers. Scenario (1) in Fig. 5 has $N_{\mathrm{sprayers}}$=12000, each injecting $6\times10^{16}$ particles s$^{-1}$, for a total mass injection rate of 69 Tg yr$^{-1}$, achieves the same forcing (–3.7 W m$^{-2}$) as scenario (2), which has $N_{\mathrm{sprayers}}$=10$^5$, each injecting $6\times10^{15}$ particles s$^{-1}$, for a total mass injection rate of 55 Tg yr$^{-1}$. As we discuss in Sect. 5.1 particle spray rates approaching $10^{17}$ s$^{-1}$ will likely result in significant particle losses due to high concentrations in the near field of the spray system, and so we consider scenario (1) to be close to the upper end of the injection rates that are likely to be feasible. From

this, it may reasonably be concluded that if MCB were ever to be used to achieve a radiative forcing close to that needed to offset a doubling of CO2, considerably more sprayers would be needed than are assumed in the estimate from Salter et al.



(2008), where only ~4500 spray vessels were assumed. The need for a greater number of sprayers in the heuristic model is primarily because of overlapping plumes, which reduce effectiveness by introducing heterogeneity into the injected particle spatial distribution. Plume overlap is not accounted for in Salter et al. (2008), wherein each sprayer uniformly increases the particle concentration over an area of $7.7 \times 10^{10}$ m$^2$. Our assumed track area is $A_t = 5.28 \times 10^{10}$ m$^2$ (Sect. 2.5) is quite similar to this, but our plumes overlap. The effect of plume overlap is demonstrated by noting that scenario (1), with fewer sprayers than scenario (2), requires ~25% more injected mass to achieve the same forcing. If we set $N_{sprayers}$ to the number that would cover the ocean if there were no overlaps (5300, Sect. 2.5), the heuristic model would require over twice as much mass to produce a forcing sufficient to offset 2xCO$_2$ as in the $N_{sprayers}=10^5$ case because the track coverage (fraction of the seeded area with at least one overlapping track) in seeded areas only marginally exceeds 50% (Fig. 5b, dashed lines). Thus, almost half of the eligible ocean area remains unperturbed in this case, requiring increases in $N_d$ to offset doubled CO$_2$ in the perturbed clouds that are harder to achieve (see Fig. 1). Figure 6 (black curves) shows $-\Delta F$ for different values of $N_{sprayers}$ plotted as a function of the total salt mass injection rate to further illustrate the need for a high number of sprayers to minimize the total mass of salt that needs to be injected to achieve a given forcing.

A key result from the heuristic model, for $D_s \sim 100$ nm, is that the forcing to offset doubled CO$_2$ should be achievable with a total salt mass injection rate of ~50-70 Tg yr$^{-1}$. This is much lower than the natural sea salt flux, which studies suggest ranges from 3,000 to >10,000 Tg yr$^{-1}$ (Textor et al., 2006; Grythe et al., 2014). The residence time of natural sea spray particles is considerably shorter than the lifetime ($\tau_{res}=2$ days) of the injected salt particles. Thus, a perhaps more useful comparison is to examine the mass loading of the injected salt particles, which increases from 0.1 to 1 μg m$^{-3}$ as forcing magnitude increases from 1 to 4 W m$^{-2}$ (Fig. 5c). The coarse mode aerosol assumed in the model (Table 1) has a mass loading of 12.7 μg m$^{-3}$, which is broadly representative of typical salt loadings in the MBL (5-20 μg m$^{-3}$, Jaeglé et al. 2011). Thus, the mass loading of injected particles required to deliver a significant radiative forcing is a relatively small fraction (~10%) of the natural salt burden in the atmosphere. This is not the case with existing climate model studies of MCB, where much higher salt mass injection rates have been required in order to provide a globally-significant radiative forcing. The reasons for this are explored the next Sect..

## 4.2 Impacts of variations in injected aerosol size and lifetime, and background aerosol concentrations

A key unresolved question concerns what size of injected particles produces the most effective brightening. Prior studies using LES and climate models have used relatively large particles with modal dry diameters exceeding 200 nm. Although particles of this size serve as very effective CCN, it is important to take into consideration the overall mass injection rate, which determines the energetic requirements for particle generation and also impacts on atmospheric chemistry (see introduction). For the sprayer number and injection rate from Scenario 2 (previous section, Fig. 5) the optimal geometric mean diameter $D_s$ of injected particles is 30-60 nm (Fig. 7a). This optimal size range is consistent with the parcel modelling of Connolly et al.





(2014) and is relatively insensitive to the background accumulation mode aerosol concentration $N_{0,acc}$. For fixed mass injection rate, injected aerosol concentration increases as the inverse third power of $D_s$ (Eqns. 5 and 7). For large injected particles ($D_s\sim200$ nm), most of the injected particles are activated (Fig. 7b), but each particle has a large mass, and so the overall mass injection rate required to produce a given forcing is roughly five times higher with $D_s$=200 nm than it is with $D_s$=100 nm (Fig. 6a), quantitatively consistent with the GCM sensitivity tests in Partanen et al. (2012). We find that 40% more forcing can be achieved per mass injected if $D_s$ is further decreased from 100 nm to 50 nm (Fig. 7a). With $D_s$ in this optimal range in

terms of forcing per mass injected, although the activated fraction is quite low, the gain in the added aerosol concentration counters this. This occurs up to a point where competition for vapor draws down supersaturation and there is a reduction in the number of injected particles that have critical supersaturations sufficiently low to activate. When $D_s$ is smaller than $\sim40$ nm, $N_d$ begins to decrease again (Fig. 7b). The saturation effect of very adding small particles is also demonstrated by the gray lines in Fig. 6a, which show forcing as function of total salt mass injection rate for $D_s$=50 nm. For low mass injection rates

($<10$ Tg yr$^{-1}$), these very small injected particles produce twice as much brightening as particles with $D_s$=100 nm. At higher rates, exceeding $\sim50$ Tg yr$^{-1}$, brightening increases very modestly.

We find a major discrepancy between the parcel model activation used here and that estimated using the ARG parameterization. For $D_s$ larger than 100 nm, droplet concentrations from ARG are in general agreement with those from the parcel model

(Fig. 7b), but as the injected particle size decreases, ARG is unable to activate a sufficient number of droplets. A significant tendency to underpredict activation fraction has been noted in several prior studies (Ghan et al., 2011; Connolly et al., 2014), and Simpson et al. (2014) found a systematic underprediction of peak supersaturations estimated with ARG, which we confirmed with the parcel model (Fig. 7c). This upshot of this issue is that whereas we find that a maximum in brightening for $D_s$ = 40 nm, the competition for vapor in ARG is so strong that it prevents activation of almost all injected particles, and so the

forcing is close to zero (Fig. 7a). It will therefore be very important to ensure that activation schemes used in climate modelling for MCB are sufficiently accurate to represent the unusual size distributions that would be needed for effective implementation of MCB.

Alterskjaer and Kristjánsson (2013, henceforth AK13) used a climate model with ARG as its activation scheme and found that

injected Aitken mode particles ($D_s$ = 44 nm) produced a strong negative $\Delta F$ at the lowest injection rate used (48.2 Tg yr$^{-1}$), but a positive $\Delta F$ for injection rates exceeding this. The behavior is consistent with our findings using the ARG scheme but is not consistent with the results from the parcel model, where brightening continues to increase with injected mass for particles of this size (Fig. 7a) and there is no cloud darkening (positive $\Delta F$). AK13 also conducted a sensitivity study in which peak supersaturations were fixed at 0.2% and found that the sign of the forcing changed from weakly positive to strongly negative.

We find that the suppression of peak supersaturation as $D_s$ decreases is similarly strong in the parcel model and the ARG parameterization for $D_s$ > 30 nm (Fig. 7c), implying that additional competition for vapor from the injected particles is not the





main reason for the reduced activation fraction in ARG. Instead, it is the general underprediction of peak supersaturation in ARG occurring at all values of $D_s$ that is the main reason for its inability to activate small Aitken particles. Indeed, the fixed supersaturation of 0.2% in the AK13 sensitivity test is quite similar to that in the parcel model (Fig. 7c), and much higher than
that for the ARG parameterization, showing that the use of ARG in AK13 is leading to misleading results regarding the efficacy of injecting Aitken mode particles.

Although the optimal geometric mean diameter $D_s$ of injected particles is 30-60 nm (Fig. 7a), there are other reasons against injecting very small Aitken-sized particles, including near-field coagulation and a higher loss rate to Brownian scavenging by
cloud droplets. The former is explored in Sect. 5.1. The timescale for Brownian scavenging of injected aerosol in a cloud-topped MBL scales with $D_s^2$ (Seinfeld and Pandis, 2003) and also decreases inversely with $N_d^{2/3}$. For realistic liquid water contents, it can be shown that under MCB (i.e., $N_d$ of a few hundred cm$^{-3}$) for $D_s$ = 60 nm, the timescale for particle losses to Brownian scavenging by cloud droplets is ~4 days, but falls to only ~1 day for $D_s$ = 30 nm.

There is strong sensitivity of forcing to injected particle residence time $\tau_{res}$ (Fig. 6b). A longer $\tau_{res}$ increases the area of each sprayer track in proportion to $\tau_{res}^2$ (since both track width and length are proportional to $\tau_{res}$, see Sect. 2.4) but the injected particle concentration over that area scales as $\tau_{res}^{-1}$ (Eqn. 7). For the scenario of many overlapping tracks ($N_{sprayers}=10^5$), the total salt mass required to produce a given forcing scales with $\tau_{res}^{-1}$ (Fig. 6b). Thus, the lifetime of injected particles is a key determinant of MCB efficacy, warranting further study.


Higher background droplet concentration lowers a cloud's albedo susceptibility ($d\alpha_c/dN_d$, Twomey 1977; Platnick and Twomey 1994). In addition, the increase in droplet concentration $\Delta N_d$ with injection is also reduced when $N_{0,acc}$ is higher (Fig. 7b). This occurs because peak supersaturation is reduced when the background particles are more numerous, and so a lower fraction of the injected aerosol is activated. This results in a forcing that scales more weakly with $N_{0,acc}$ than would be
expected based solely on the albedo susceptibility. As $N_{0,acc}$ increases from 50 to 150 cm$^{-3}$, the albedo susceptibility decreases by a factor of three, and yet the magnitude of the radiative forcing (e.g., for $D_s$=50 nm) decreases by less than a factor of two.

The presence of a coarse mode in the unperturbed state imposes a relatively modest decrease in the effectiveness of aerosol injections in brightening clouds. For $D_s$ in the range 50-100 nm, the realistic background coarse mode concentration
($N_{0,coarse}=10$ cm$^{-3}$) used throughout this study results in a forcing that is less than 10% smaller than in the absence of a coarse mode (Fig. 8), and the effect is weaker for larger $D_s$. Coarse mode concentrations vary with wind speed, and can reach values of ~20-50 cm$^{-3}$ at high wind speeds (Zheng et al., 2018). As Fig. 8 shows, it is only at concentrations well in excess of 10 cm$^{-3}$ (mass loadings well in excess of 10 $\mu$g m$^{-3}$) that there would be significant limits to brightening due to the coarse mode. It is important that MCB spray technology does not introduce a significant number of coarse mode particles, as these will reduce



brightening. However, it should be noted that the coarse mode mass loadings required to produce a significant dampening of
forcing are considerably in excess of those required to produce brightening using ~100 nm particles.

### 4.3 Sensitivity to updraft speed

Updraft speed $w$ is a key determinant of the peak supersaturation during the activation process in updrafts (Sect. 2.6). A single
value ($w$=0.4 m s$^{-1}$) is assumed for the results shown in this study, but note that the sensitivity of $\Delta F$ to $w$ is strongly dependent

upon the injected particle size (Fig. 9a), with sensitivity decreasing strongly as $D_s$ increases. The sensitivity is related to $N_d$
(Fig. 9b), which itself depends upon the peak supersaturation in the updraft (Fig. 9c) and the size distribution of injected and
unperturbed aerosol particles. Note that the suppression in peak supersaturation for the perturbed case compared with the
unperturbed case is stronger for $D_s = 100$ nm than it is for $D_s = 200$ nm, but falls no further for $D_s = 50$ nm. Smaller and more
numerous particles have a greater surface area and therefore remove vapor more rapidly, but kinetic limitations on growth

rates restrict the continuation of this when $D_s$ falls much below 100 nm. For the mass spray rates assumed here (scenario 2,
see Sect. 4.1), almost all injected particles activate in updrafts exceeding 0.3 m s$^{-1}$ when $D_s = 200$ nm (Fig. 9b). The
suppression of peak supersaturation is relatively modest in this case (Fig. 9c). For $D_s = 100$ nm the forcing magnitude increase
with $w$ is stronger because the $N_d'$ increase with $w$ is stronger. However, it should be noted that for $D_s$=100 nm, the forcing
magnitude only increases by 30% over the range $0.2 < w < 0.6$ m s$^{-1}$, indicating relatively weak sensitivity to updraft speed

overall. The greatest sensitivity to $w$ occurs for the smallest injected particles ($D_s$=50 nm) where the forcing increases by 80%
over the range $0.2 < w < 0.6$ m s$^{-1}$. This reflects the fact that the greatest sensitivity of $N_d'$ to increasing peak supersaturation
will occur when the critical supersaturation of the modal diameter (where the greatest number of particles lies) is close to the
peak supersaturation. A 50 nm diameter salt particle has a critical supersaturation of 0.25% (e.g., Petters and Kreidenweis
2007), which is similar to the peak supersaturation at $w = 0.4$ m s$^{-1}$ (Fig. 9c). Given that activation in real MBL clouds occurs

in a spectrum of updrafts (e.g., Snider et al., 2003), this result would caution against the use of injected particles that are too
small to increase $N_d$ in the majority of clouds seeded.

### 4.4 Direct radiative forcing

A recent study with three climate models found that direct forcing from injected salt aerosol may compete with or even exceed
the indirect forcing in magnitude (Ahlm et al., 2017). We demonstrated (Sect. 4.2) that injected particles with sizes

$D_s$=30-60 nm produce the greatest brightening for a given salt mass injected, and a forcing to offset doubled $CO_2$ can be
achieved with injection rates below 100 Tg yr$^{-1}$ (dashed circle, Fig. 10a). These "optimal" particle sizes are much smaller than
the dry modal diameters of 200, 260 and 880 nm for injected particles in the models used for the GeoMIP assessment of Ahlm
et al. (2017). The magnitude of the heuristic model global direct forcing $\Delta F_{direct}$ (Sect. 2.7) is very small (<0.5 W m$^{-2}$) for total
salt mass injection rates of <100 Tg yr$^{-1}$ (Fig 10b). Indeed, to generate $\Delta F_{direct} = -4$ W m$^{-2}$ requires an order of magnitude

greater mass injection rate than it does to produce the same forcing from MCB (compare Fig. 10a and 10b). Partanen et al.





(2012) found very small contribution of direct radiative forcing with $D_s$ = 100 nm, because the same indirect forcing in this case was achieved with ~5 times less injected mass than their case ("5×GEO") with $D_s$=200 nm, wherein direct forcing constituted about 30% of the forcing.

The dry size $D_s$ to maximize the direct forcing, for a given mass injection rate, is ~110 nm (Fig. 10b), which is around twice as large as the optimal size for MCB. As we have seen (Figs. 6 and 7), producing significant cloud brightening for the $D_s$ values in the models in Ahlm et al. (2017) requires much higher mass injection, and this leads to significant direct radiative forcing. Indeed, for $D_s$=880 nm, the brightening efficiency (Fig. 7) is so low that we would expect very little brightening for spray rates of several hundred Tg yr$^{-1}$, which is consistent with the very small increases in $N_d$ for the model that injected

particles of this size, despite an injection rate of 590 Tg yr$^{-1}$. In conclusion, the results here demonstrate that *marine cloud brightening is not very effective without clouds* when consideration is given to the injection rates required to produce a significant radiative forcing.

## 5 Implications for future work to test marine cloud brightening

The heuristic model results presented here, together with the assessment of LES studies, have implications that may help guide

future work to test the concept of MCB to cool Earth. Broadly speaking, these implications fall into three categories: guidance for the engineering development of particle injection (sprayer) systems; guidance for the design of climate model simulations to evaluate the feasibility of regional and global marine cloud brightening; suggestions for future LES modelling.

### 5.1 Sprayer development considerations

The results presented in Sect. 4.1 suggest that in order to produce global radiative forcing from MCB that offsets a significant

fraction of the forcing from doubling $CO_2$, a large number of sprayers will be required. In order to keep the number to below ~$10^5$, particle number injection rates $\dot{N}_s$ of ~$10^{16}$ s$^{-1}$ will be needed. Similar forcing can be achieved with fewer sprayers, but this will necessitate higher $\dot{N}_s$. This implies very high particle concentrations in the near-field of the spray system. Taking the spray system to be a collection of nozzles arranged over some area $A_0$, spraying into an airflow $v_{flow}$, then the initial particle concentration $N_0$ in the immediate wake of the sprayer is $N_0 = \dot{N}_s/A_0 v_{flow}$. The approach in Turco and Yu (1997) is used to

model the downstream particle concentration assuming a fixed coagulation kernel based on a particle diameter $d$ = 100 nm and $v_{flow}$=10 m s$^{-1}$. The coagulation kernel is not strongly dependent upon particle diameter for this size range (see Fig. 13.5 in Seinfeld and Pandis, 2003), so variations in injected dry size and the degree of hygroscopic swelling do not have major impacts. We consider a diluting slender plume based on the Gaussian plume dispersion which yields a plume cross sectional area (and thus volume) that evolves with time $t$ as $V = V_0 \left(1 + \left[\frac{t}{t_{dil}}\right]^a\right)$. Here $a = r_y + r_z$ =1.49 and $t_{dil}$ =1.2 s, with

$t_{dil}=v_{flow}^{-1}\left(A_0/\pi R_y R_z\right)^{1/(r_y+r_z)}$, where the plume widths at distance $x$ downstream of the sprayer in the cross-wind and vertical



directions are $\sigma_y(x) = R_y x^{r_y}$ and $\sigma_z(x) = R_z x^{r_z}$ respectively. The constants $R_y$, $r_y$, $R_z$ and $r_z$ are those for neutral stability conditions from Klug (1969) as reproduced in Table 18.3 of Seinfeld and Pandis (2003). Numerical simulations had to be performed because the solution does not allow for analytical integration.

Results of the coagulation-dilution calculations (Fig. 11a) indicate that there are relatively weak particle losses from coagulation until particle injection rates $\dot{N}_s$ exceed ~$10^{16}$ s$^{-1}$, above which loss rates grow sharply. Without dilution there are large losses within the first 100 s for rates exceeding $10^{15}$ s$^{-1}$, and for rates approaching $10^{17}$ s$^{-1}$ most of the losses occur within the first 10 seconds. Dilution immediately downstream of the spray system is therefore most important for particle survival. The volume profile for these simulations is shown in Fig. 11b. We assume that particle concentration within the expanding

plume is uniform, which is somewhat unrealistic, because the edges of the plume will become more dilute at a faster rate than those in the center. A multi-shelled Gaussian plume model was employed in Stuart et al. (2013) and this appears to result in somewhat weaker loss rates than we find, but the same general dependencies were found. There is a somewhat weaker dependence of the fraction of particles remaining on $A_0$ than might be imagined (Fig. 11c), given that $A_0$ determines the initial concentration of particles, and loss rates scale with $N^2$. This is because it takes longer for turbulent eddies to penetrate into a

wider plume and mix ambient air into the plume core, so that larger $A_0$ is associated with a longer dilution timescale $t_{dil}$.

The strong dependence of coagulation losses on $\dot{N}_s$ (Fig. 11d) indicates that as rates approach $10^{17}$ s$^{-1}$ the fraction of particles remaining decreases as rapidly as $\dot{N}_s$ increases, which essentially means no increase in the far-field concentration as injection rates are further increased. Because this is not strongly sensitive to $A_0$ (Fig. 11c), this imposes a fairly hard limit (~$5\times10^{16}$ s$^{-1}$)

on the maximum rate of particles that a ship-deployable spray system can provide to the far-field environment. Recall that the spray scenario to offset doubled $CO_2$ forcing with 15000 ships discussed in Sect. 4.1 requires a number injection rate that is at this upper limit of feasibility. More rapid dilution than can be provided by boundary layer turbulence may be possible in the first few seconds downstream of the spray system if the initial flow rate can be increased, but more sophisticated fluid and aerosol dynamics modelling will be required to determine the maximum far-field injection rates that a sprayer can provide.

**5.2 Climate modelling**

The results of this study have implications for both types of climate modelling MCB studies discussed in the introduction, namely those that fix $N_d$ at some value in seeded regions and those that attempt to model aerosol-cloud interactions using salt aerosol injection.

The results presented in Sect. 4.1 demonstrate that there are diminishing returns on increasing $N_d$ as spray rate increases (e.g., Fig. 5b). Producing cloud droplet concentrations of 1000 cm$^{-3}$ is possible but requires mass injection rates approaching 1000 Tg yr$^{-1}$ (compare Fig. 5a and 5b). Locally high mass injection rates would reduce the fractional area of the





ocean required for spraying (see next paragraph), but it should be borne in mind that increasing $N_d$ to 1000 cm$^{-3}$, as has been done in some climate modelling studies (e.g., Rasch et al., 2009; Baughman et al., 2012), would increase the Brownian

scavenging rate of interstitial injected aerosol and reduce the overall particle residence time $\tau_{res}$, resulting in a reduced forcing (see Sect. 4.2, Fig. 6b). Increasing $N_d$ from 300 to 1000 cm$^{-3}$ reduces the Brownian scavenging timescale by a factor of over 2 (Seinfeld and Pandis 2003), suggesting that attempting to implement MCB with very high droplet concentrations may not be practical.

Several previous climate model studies have seeded a relatively small fraction of the ocean area. Jones et al. (2009) set $N_d' = 375$ cm$^{-3}$ in three regions totalling only 4.7% of the ocean and achieved $\Delta F = -0.97$ W m$^{-2}$. The unperturbed $N_d$ is not known for this study so it is not possible to predict the Twomey forcing for this case using the heuristic model, but for $f_{spray} = 0.1$, a peak forcing of -1.4 W m$^{-2}$ is achievable (Fig. 12) but achieving a forcing magnitude in excess of 1 W m$^{-2}$ requires a mean $N_d$ of over 800 cm$^{-3}$ ($r_N \sim 8$) in the seeded area. One can only conclude that either the unperturbed $N_d$ in Jones

et al. (2009) was very low, or that the model produced significant positive cloud adjustments that augmented the Twomey effect. Fig. 12 demonstrates that global forcing magnitudes greater than ~4 W m$^{-2}$ from the Twomey effect alone are only likely to be possible if ~50% of the eligible ocean area (~40% of the total ocean area) is seeded.

Climate modelling to investigate MCB by injecting surface salt sources has typically injected particles with $D_s$ of 200 nm or

greater (Alterskjaer et al. 2012, 2013; Ahlm et al., 2017). The heuristic model sensitivity to injected particle size presented in Sect. 4.2 indicates that $D_s = 200$ nm is inefficient (Fig. 7), requiring mass spray rates 5 times higher than for $D_s = 100$ nm and over an order of magnitude higher than for $D_s = 50$ nm (Fig. 6). This has led to mass spray rates in existing MCB studies of hundreds of Tg yr$^{-1}$ (Ahlm et al., 2017), and these high mass spray rates produce significant direct radiative forcing (Ahlm et al., 2017). Our results suggest that smaller injected particles can yield global radiative forcing of -1 to -4 W m$^{-2}$ with very little

direct radiative forcing. Providing meaningful global radiative forcing using the aerosol direct effect is an extremely inefficient use of salt particles. We therefore suggest that future climate modelling should focus on smaller injected particles, to build on the sensitivity study in Partanen et al. (2012). This may challenge some models, because a dedicated injection particle mode independent of the model's accumulation mode will be required, and accurate treatment of activation for such cases is a major issue (see Sect. 4.2 and the Appendix).

**5.3 Challenges for LES modelling**

One notable feature of existing LES studies designed to test the sensitivity of albedo to particle injections (Sect. 3) is that all existing experiments show some degree of brightening, although some do show cloud cover or condensate adjustments that partly offset the Twomey effect. In most cases, especially very clean conditions, cloud adjustments significantly augment Twomey brightening. No LES studies in the literature show domain-wide cloud adjustments that completely offset Twomey





brightening. However, predicting cloud adjustments accurately is a challenge, even for LES models; some models do not represent physical processes needed to produce the correct adjustments. Such processes include size-dependent droplet evaporation, which requires an estimation of supersaturation (many LES assume saturation adjustment; see Wang et al., 2003) and sedimentation-entrainment feedback (e.g., Bretherton et al., 2007). Tests to establish the importance of these processes for determining susceptibility to particle injections are incomplete. In addition, existing LES studies represent only a small subset

of possible meteorological conditions, focusing primarily upon shallow MBLs that are probably more susceptible to aerosol injections. Studies examining the susceptibility to particle injections of deeper trade wind MBLs including aggregated shallow convective systems need to be conducted to establish the efficacy of MCB in these regions.

It should be noted that most LES cases to test MCB are insufficiently short to examine the responses at timescales longer than

the injected particle residence time $\tau_{\mathrm{res}}$, which suggests that longer simulations will be necessary to evaluate the true radiative forcing from injections. Further, no LES studies to date have attempted to constrain the injected particle residence time $\tau_{\mathrm{res}}$, which is a key determinant of MCB forcing (Sect. 4.2, Fig. 6b). The dependence of $\tau_{\mathrm{res}}$ on precipitation, entrainment, in-cloud Brownian scavenging, and other factors, warrants exploration using LES. In addition, there are a number of potentially important feedbacks involving aerosol residence time that are ideal for study using LES. First, it is well understood that MBL

precipitation is a major sink of CCN (Wood et al., 2012; Zheng et al., 2018; Wang et al., 2021). Residence time is expected to be shorter in precipitating MBLs, and suppression of precipitation by high $N_{\mathrm{d}}$ in seeded clouds will increase $\tau_{\mathrm{res}}$ (Wood 2012), but increased cloud surface area will increase Brownian scavenging of injected particles, reducing $\tau_{\mathrm{res}}$. Precipitation suppression will also impact the background aerosol properties, including potentially increasing the concentration of coarse mode and giant CCN that may counter some of the $N_{\mathrm{d}}$-driven precipitation suppression (Feingold et al., 1999). These feedback

processes may affect aerosol residence time and MCB forcing in ways not accounted for in current analyses. Finally, deficiencies in some activation schemes used in LES are likely to be a significant issue hindering accurate representation of the effects of injected particles smaller than 100 nm (Sect. 4.2, Appendix, and Connolly et al., 2014), so it will be important to ensure that LES models can handle the activation process faithfully.

## 6 Conclusions

This study presented a simple heuristic model to produce useful quantitative estimates of the radiative forcing from the Twomey effect driven by salt particle injections over the global oceans (marine cloud brightening, MCB). The model includes a treatment of individual sprayer plumes and their overlap, and so can be used to explore brightening as a function of the number of sprayers. Brightening is predicted using Twomey's albedo susceptibility given predicted increases in $N_{\mathrm{d}}$ from an activation look-up table derived using Lagrangian parcel modeling that incorporates both background and injected aerosol

particle size distributions. Parameters for the model are constrained with observations of cloud cover and $N_{\mathrm{d}}$ from satellites, along with aerosol properties from syntheses of in situ observations. The model performs reasonably well in estimating the





cloud brightening from a number of large eddy simulations (LES) reported in the literature, although the LES cases tend to produce more (less) brightening for clean (polluted) cases, likely because of cloud adjustments (changes in cloud cover and/or liquid water path in response to aerosols) that are not included in the heuristic model.


The heuristic model is then used to estimate global radiative forcing from MCB and its sensitivity to injected particle spray rates and particle sizes. The key conclusions of the work are:

- Radiative forcing to offset doubled $CO_2$ can be achieved with global mean salt spray rates of ~50-70 Tg yr$^{-1}$. This is
much lower than the natural sea salt flux, and much lower than spray rates used in global models, which have injected larger particles than are needed to efficiently brighten clouds. To produce this radiative forcing, a large number of sprayers ($10^4$-$10^5$) will be required to operate over the majority of the 54% of the Earth's surface that is over ocean and remote from land.
- Injected particles with geometric mean dry diameters of 30-60 nm are most efficient at brightening clouds for a
fixed mass of salt injected.
- There is no evidence for marine cloud darkening (positive radiative forcing) using the parcel model-based activation scheme, although the Abdul Razzak and Ghan (2000) parameterization incorrectly shows that this occurs for very high concentrations of small injected particles due to excessive competition for water vapor.
- Competition for vapor effectively limits the maximum possible magnitude of radiative forcing from MCB to
approximately 8 W m$^{-2}$ for salt spray rates less than 1000 Tg yr$^{-1}$, assuming all ocean regions are seeded. This assumes that negative cloud adjustments remain relatively small compared with the Twomey effect.
- Heuristic model radiative forcing estimates are mostly within a factor of 3 of those from LES, across a range of different spray and unperturbed conditions.
- Brightening in the heuristic model and the LES decreases strongly with the aerosol/droplet concentrations in the
unperturbed clouds, so it is critical to better understand and model the seasonal and geographical variations in these parameters in order to identify optimal locations and times for particle injections and to predict radiative forcing.
- For injected particles with geometric mean dry diameters of ~100 nm or more, there is relatively weak sensitivity to updraft speed for values larger than 0.2 m s$^{-1}$.

Direct radiative forcing from injected particles is very small for mass injection rates less than 100 Tg yr$^{-1}$. MCB is
far less effective without clouds when consideration is given to the quantity of salt that must be injected.





## Appendix: Parcel model emulator

To determine aerosol activation, an explicit Lagrangian parcel model is used to construct a five-dimensional look-up table that predicts peak supersaturation and the concentration of activated aerosol (cloud droplet concentration $N_d$) as a function of updraft speed $w$, the concentration $N_s$ and geometric mean dry diameter $c$ of the injected particles respectively, and the unperturbed particle concentrations of accumulation mode $N_{0,acc}$ and coarse mode $N_{0,coarse}$ particles. All other aerosol size and hygroscopicity parameters are fixed at their values in Table 1. The temperature and pressure are fixed at 280 K and 900 hPa respectively. The Lagrangian parcel model is initialized just below cloud base with a saturation ratio of 0.99 and is integrated to a height of 50 m above the saturation level. Particles are determined to be activated if they have reached a diameter of 2 μm at this height. Split timestepping is used in order to render the integration stable for the very small sizes of some of the injected particles. The discretization of the dry size distribution is set for each case in order to provide an accurate estimate of the number of activated droplets. Parcel model simulations are produced for all 6468 permutations of the values of the five input parameters shown in Table A1. The injected particle concentrations $N_s$ are set to be a function of $D_s$ such that the overall mass of injected particles is the same for a given value of the scaling factor (Table A1). A scaling factor of unity corresponds to an injected mode mass loading of 1.2 μg mg⁻¹. For the look-up table, basic linear interpolation in five dimensions is used to determine the droplet concentration and peak supersaturation for any given set of input variables $w$, $D_s$, $N_s$, $N_{0,acc}$, and $N_{0,coarse}$.

Figure A1 provides a comparison between the parcel model results and the ARG parameterization. In general, ARG significantly underpredicts the peak supersaturation (Fig. A1 panels a, c and e) in all cases. Despite this, there is good agreement between the parcel model and ARG droplet concentrations for $D_s$>200 nm. For this size range, the injected particles have critical supersaturations small enough that despite the underprediction of peak supersaturation in ARG, it is sufficiently high to activate most injected particles. However, as $D_s$ falls below 200 nm, the underprediction of peak supersaturation in ARG has increasingly severe consequences, and this is exacerbated at the highest mass loadings. For $D_s$= 50 nm, ARG $N_d$ is only around 50% of that in the parcel model, and this underprediction falls rapidly as $D_s$ falls further. Based on these findings, we conclude that the biases in ARG are too large for this parameterization to produce useful results.

## Code and Data availability

All code and data used in this study are available on request from the author.



## Competing interests

The authors declare that they have no conflict of interest.

## Author contribution

The sole author Robert Wood conceived and carried out the work presented, and wrote the manuscript.

## Acknowledgements

Support for this work was provided by the National Oceanographic and Atmospheric Administration (NOAA Award NA20OAR4320271), and from Lowercarbon, the Pritzker Innovation Fund and SilverLining through the Marine Cloud Brightening Project. LES simulations were performed at the University of Washington by Peter Blossey and Je-Yun Chun, with assistance from Matthew Wyant. Sarah Doherty, Phil Rasch, Kelly Wanser, Tom Ackerman, Peter Blossey, Matthew Wyant, Ehsan Erfani, Je-Yun Chun, Armand Neukermans, Chris Bretherton, Gary Cooper, Sean Garner, Kate Murphy, Paul Connolly, and Michael Diamond are thanked for discussions that have helped to frame and improve this work.

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





**Table 1: Parameters used in the heuristic model and their assumed values**

| Symbol | Parameter | Assumed value(s) | Justification |
|---|---|---|---|
| $\alpha_c$ | Unperturbed cloud albedo | 0.56 | Bender et al. (2011). See text. |
| $\phi_{atm}$ | Atmospheric correction factor | 0.70 | Based on Diamond et al. (2020). |
| $f_{ocean}$ | Fraction of Earth's surface covered by ocean eligible for spraying | 0.54 | Divide globe into $10 \times 10°$ boxes (approximate sprayer plume length). Only boxes with less than 10% land eligible for spraying to minimize plumes intercepting land areas. See text. |
| $f_{spray}$ | Fraction of eligible ocean areas in which sprayers operate | 0.5-1.0 | |
| $f_{low}$ | Fraction of sprayed area covered by stratiform low clouds unobscured by high clouds | Function of $f_{spray}$ decreasing from 0.68 for $f_{spray}<0.1$ to 0.33 for $f_{spray}=1$ | Uses MODIS liquid cloud fraction. For $f_{spray}<1$, sort eligible $10 \times 10°$ boxes by their monthly climatological mean liquid cloud fraction. Set $f_{low}$ equal to the mean of the cloudiest $f_{spray}$ fraction. See Equation 4 in Sect. 2.2 |
| $F_\odot$ | Solar irradiance | 342 W m$^{-2}$ | Assumed day+night averaged global mean solar irradiance. |
| $\dot{M}_s$ | Rate of NaCl injection by each sprayer | 1-1000 kg hr$^{-1}$ | Variable |
| $N_{sprayers}$ | Number of sprayer vessels deployed | $3\times10^3$-$3\times10^5$ | Variable |
| $D_s$ | Geometric mean diameter of injected NaCl particles | 10-1000 nm | Variable |
| $S$ | Geometric standard dev. of injected NaCl particle size | 1.6 | |
| $\tau_{res}$ | Residence time of injected particles | 2 days | Based on Wood et al. (2012) |
| $D_{0,acc}$ $D_{0,coarse}$ | Geometric mean diameter of background aerosol | 175 nm (acc.) 615 nm (coarse) | Accumulation mode size values based on marine aerosol climatology of Heintzenberg et al. (2000). Coarse mode values from Zheng et al. (2018). |
| $S_{0,acc}$ $S_{0,coarse}$ | Geometric standard dev. of background aerosol size | 1.5 (acc.) 1.8 (coarse) | |
| $N_{0,acc}$ $N_{0,coarse}$ | Number concentration of background aerosol | 50-150 cm$^{-3}$ (acc.) 10 cm$^{-3}$ (coarse) | Coarse mode value from summer mean at the Graciosa Island from Zheng et al. (2018). |
| $\kappa$ | Aerosol hygroscopicity | 0.7(acc.) 1.2 (coarse, injected) | Accumulation mode from Pringle et al. (2010) Coarse mode/injected salt from Petters and Kreidenweis (2007) |
| $w$ | Updraft speed for aerosol activation | 0.4 m s$^{-1}$ | Approximate value based on numerous stratocumulus field experiments |
| $U_0$ | Mean surface wind speed | 7 m s$^{-1}$ | Mean near-surface wind over global ocean (Archer and Jacobson 2005) |
| $h$ | Marine PBL depth | 1 km | Typical mean value for marine low clouds over oceans |
| $K$ | Plume lateral spread rate | 1.85 km h$^{-1}$ | Based on observed ship track spreading rate (Durkee et al., 2000) |




**Table 2: Large eddy simulation studies included in this study. Information included in this table focused on highlighting diversity in injected particles and domain size.**

| Study | *Wang et al. (2011)* **[W11]** | *Berner et al. (2015)* **[B15]** | *Jenkins et al. (2013)* **[J13]** | *Possner et al. (2018)* **[P18]** | *Chun et al. (2020)* **[C21]** | |
|---|---|---|---|---|---|---|
| **Case info** | DYCOMS-II RF02 | Collapsed MBL (Sanko Peace) | DYCOMS-II RF02 | VOCALS RF06 Deep open cell | (a) Collapsed MBL (Sanko Peace) | (b) CGILS S12 Shallow MBL |
| **PBL depth, m** | 900 | 350 | 750 | 1500 | 350 | 700 |
| **Spray rate, # s⁻¹ [diameter, nm]** | $1.04\times10^{16}$ [200] | $1.8\times10^{15}$ (clean) $1.8\times10^{15}$ (poll.) [200] | $5.6\times10^{16}$ $1.1\times10^{16}$ (weak) [200] | $1.04\times10^{16}$ [600] | $10^{16}$ [50] | $10^{16}$ [50] |
| **Spray duration** | 30 hr | 10.7 min | 30 min | 48 hr | 7.6 min | 7.6 min |
| **Mass of NaCl emitted total, kg [per hour, kg]** | 10320 [344] | 10 [57] | 1105 [2210] | $4.5\times10^{5}$ [9400] | 0.7 [5.2] | 0.7 [5.2] |
| **Simulation duration** | 30 hr | 8 hr | 5 hr | 48 hr | 8 hr | 8 hr |
| **Domain size, km×km [area, km²]** | 60×120 [7200] | 51.2×12.8 [660] | 9×9 [81] | 180×180 [32000] | 48×9.6 [460] | 24×4.8 115 |
| **Number of particles emitted [cm⁻³]** | 174 | 4 | 1650 | 370 | 29 | 57 |
| **Spray details** | Sprayer traverses long edge of domain several times | Sprayer passes through short edge of domain once only | Sprayer traverses domain once only | Sprayer traverses domain several times | Sprayer traverses short edge of domain once only | |
| **Experiments used** | Wet and dry profiles used. Wet cases 50, 100 and 200 cm³ CCN initially. Dry case has 100 cm⁻³ CCN. Use single sprayer and uniform seeding only | Basetrack (clean) and SensHiAer (polluted) case used. | Non-precipitating (NP-Ch; NP-Pa) and Precipitating (WP) cases used. Sensitivity study with weaker sprayer on WP. | Single experiment | Two experiments, one with reduced sea spray aerosol | Single experiment |






**Table A1: Parameter values used to construct the activation look-up table**

| Variable | Name | Values used |
|---|---|---|
| $w$ | Updraft speed | 0.2, 0.4, 0.6 m s$^{-1}$ |
| $D_s$ | Injected particle geometric mean dry diameter | 15, 30, 45, 60, 80, 100, 150, 200, 300, 500, 1000 nm |
| $N_s$ | Injected particle concentration | Scaling factor of [0, 0.1, 0.3, 1, 3, 10, 30] times $400(100/D_s)^3$ mg$^{-1}$, where $D_s$ is in nm |
| $N_{0,acc}$ | Unperturbed accumulation mode concentration | 5, 10, 50, 100, 150, 200, 500 mg$^{-1}$ |
| $N_{0,coarse}$ | Unperturbed coarse mode concentration | 0, 3, 10, 30 mg$^{-1}$ |






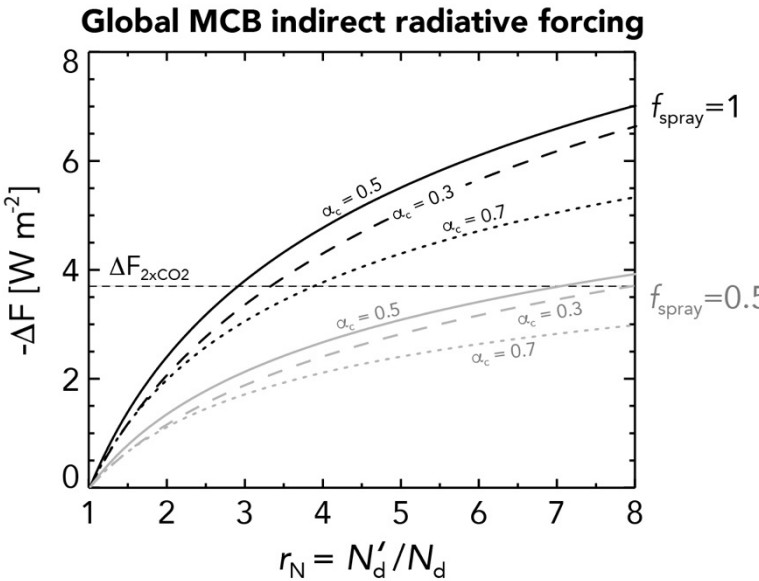

**Figure 1: Global radiative forcing from marine cloud brightening $\Delta F$ as a function of the ratio of the perturbed to unperturbed (background) cloud droplet concentration $r_N = N'_d/N_d$. Curves are shown for the case where sprayers are deployed over all eligible ocean regions ($f_{spray} = 1$, black lines) and where sprayers are deployed over only 50% of these areas ($f_{spray} = 0.5$, gray lines), for unperturbed cloud albedos $\alpha_c$ ranging from 0.3-0.7. The fraction of the Earth's surface area eligible for seeding is $f_{ocean} = 0.54$, and the atmospheric correction factor $\phi_{atm} = 0.7$**



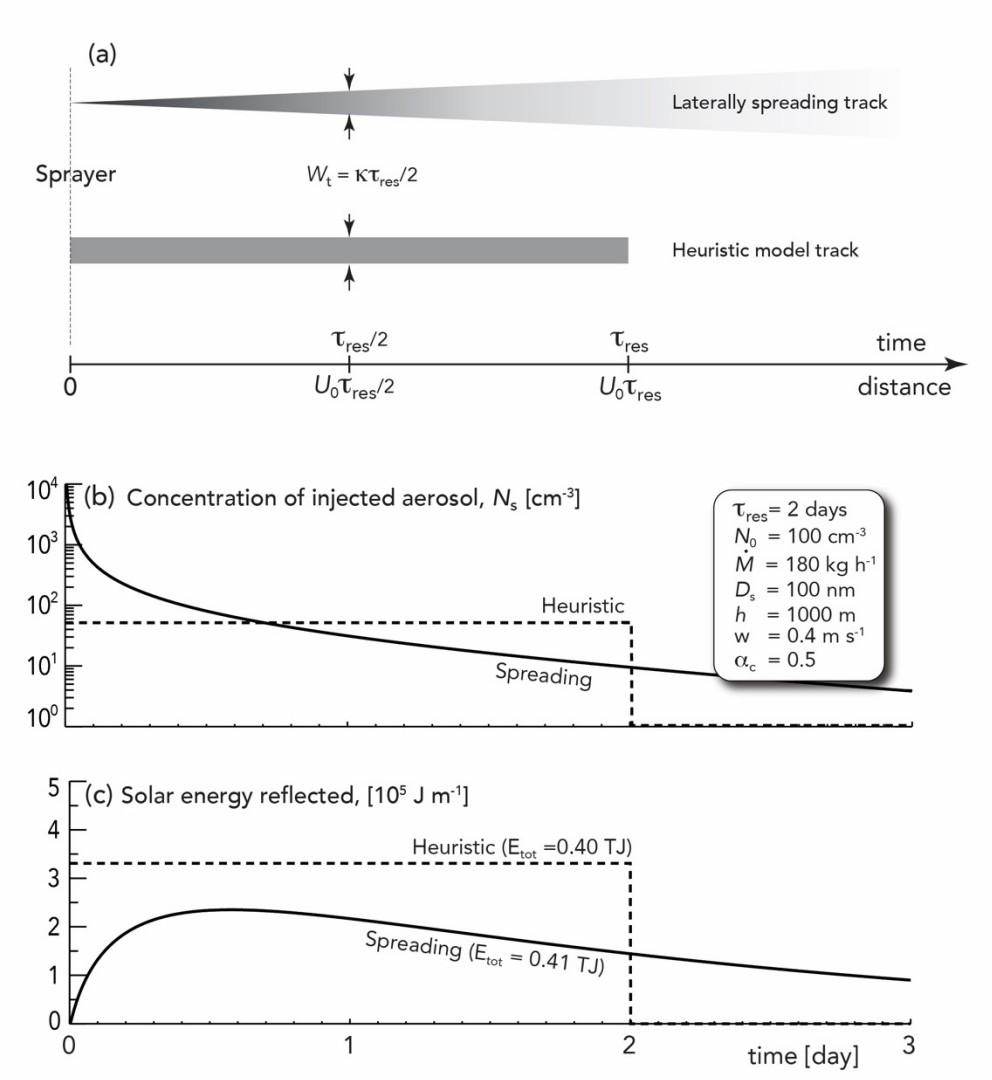

Figure 2: (a) Plan view of a realistic laterally-spreading plume/track (top) and the track assumed in the heuristic model, as a function of time/distance downstream of the sprayer. The shading qualitatively indicates the injected particle concentration. (b) Injected aerosol concentration as a function of time for the two plume types given the spray injection information in the box. A residence time $\tau_{res} = 2$ days, and a widening rate $K = 1.85$ km hr$^{-1}$ are assumed. (c) additional reflected solar radiation per meter length of track from aerosol-cloud interactions for the two plume types. The total additional reflected energy $E_{tot}$ from the two plumes is very similar. In this case, approximately 43% of the energy reflected from the spreading plume occurs for times $t > \tau_{res}$.



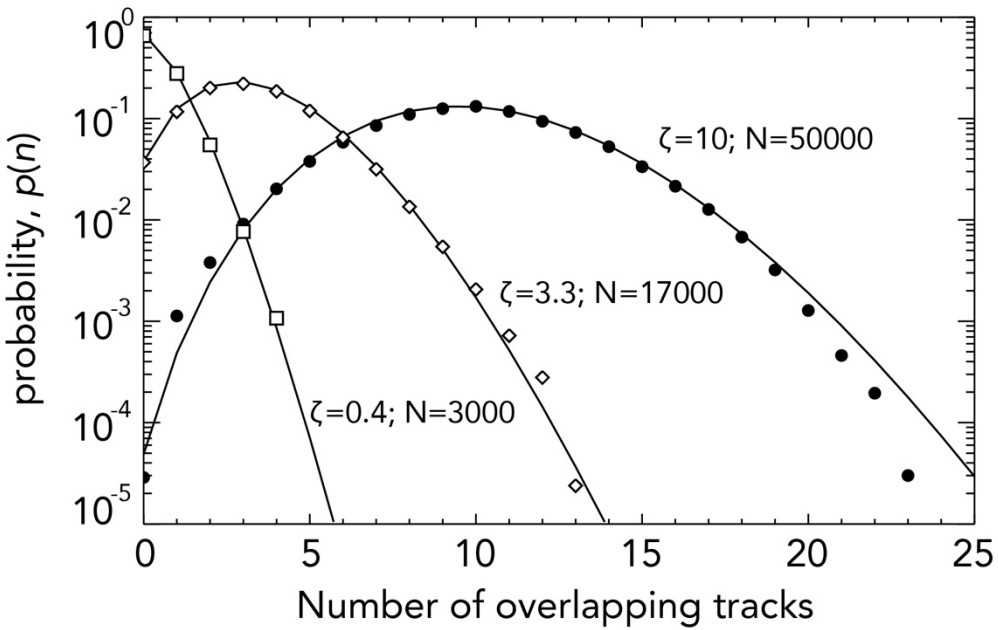

**Figure 3: Probability density functions $p(n)$ derived from the Monte Carlo simulations of overlapping rectangular tracks for three values of the mean track density $\zeta$ (0.4, 3.3 and 10). For the simulations, a domain of size 4000×4000 km is modelled using a 4000×4000 array, and tracks of length 1200 km and width 44 km (see Sect. 2.5) are placed randomly to the array, assuming periodic boundary conditions. The long dimension of each track is randomly set to be parallel to the $i$ or $j$ direction of the box. Poisson distributions (Equation 8) are shown corresponding based on the mean track densities and represent an excellent fit to the data. The track densities $\zeta$ = 0.4, 3.3 and 10 correspond to a total number of ships, if spraying were to take place over the entire eligible ocean region, of ~3000, ~17000, and ~50000 respectively.**

l175

l180

l185

l190



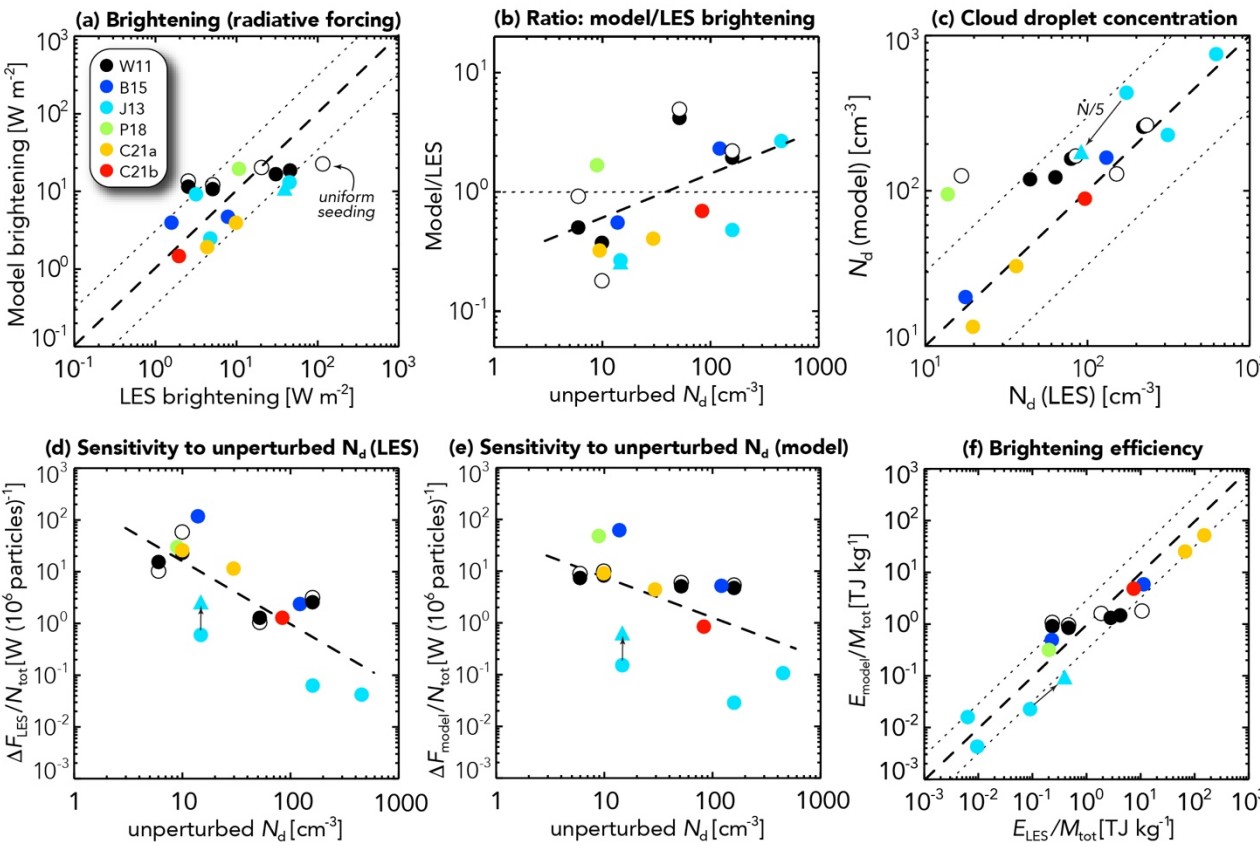


**Figure 4: Comparison of heuristic model and LES results. Each color indicates a different study (see Table 2 for details). For W11, open circles indicate cases with uniform seeding across the domain. (a) Brightening (radiative forcing) for the LES and heuristic model. Dashed line indicates agreement, and dotted lines represent factor of 3 differences; (b) Ratio of the brightening in the heuristic model to that in the LES, plotted against the droplet concentration in the unperturbed case $N_d$. The dashed line shows the linear**
**least squares fit to the data; (c) Modeled vs LES cloud droplet concentration given the injection rates and particle size distribution employed in the model (see text); Sensitivity of the normalized forcing (expressed as Watts per injected particle) to the unperturbed $N_d$ for (d) the LES experiments; (e) the heuristic model, with lines representing least-squares fits to the data; (f) the brightening efficiency expressed as the energy reflected over the course of the experiments per mass of salt injected. For J13, the triangle indicates reduced injection rate by a factor of 5, with the arrows connecting the experiments with full and reduced injection rates.**






**Figure 5: (a)** Global mean radiative forcing $\Delta F$ (colors) and total flux of sodium chloride (dotted contours) from MCB applied to all eligible ocean areas (54% of Earth's surface) as a function of the number of sprayers $N_{sprayers}$ and the salt mass injection rate $\dot{M}_s$ for each sprayer; **(b)** increase in cloud droplet concentration $\Delta N_d$ (colors), mean fraction of aerosol activated in tracks (dotted contours), and track coverage (dashed contours); **(c)** injected aerosol number concentration in tracks (colors) and mean mass loading in the MBL of injected salt (dotted contours). The inset of panel (a) shows the key model parameters, with others as in Table 1. Two scenarios (1) and (2), which each produce sufficient forcing to offset doubled $CO_2$ are highlighted. Scenario (2) has a higher number of sprayers but a lower rate of particles injected per sprayer.


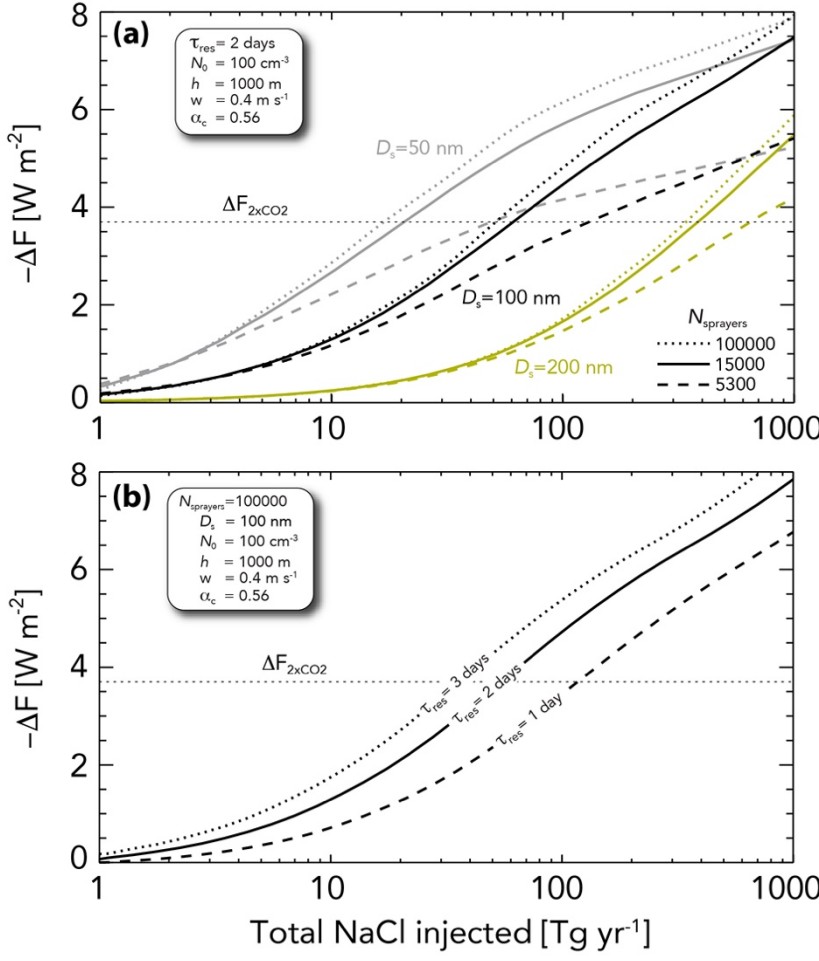

**Figure 6: Radiative forcing as a function of the total (global) rate of salt mass injection, for (a) three sprayer numbers ($N_{sprayers}$=5300, 15000, and 120000), and for geometric mean spray diameters $D_s$ of 50 nm (gray), 100 nm (black), and 200 nm (yellow); (b) forcing for $N_{sprayers}$=120000 and $D_s$=100 nm as a function of injected particle lifetime. All other parameters are the same as those used in Fig. 5.**




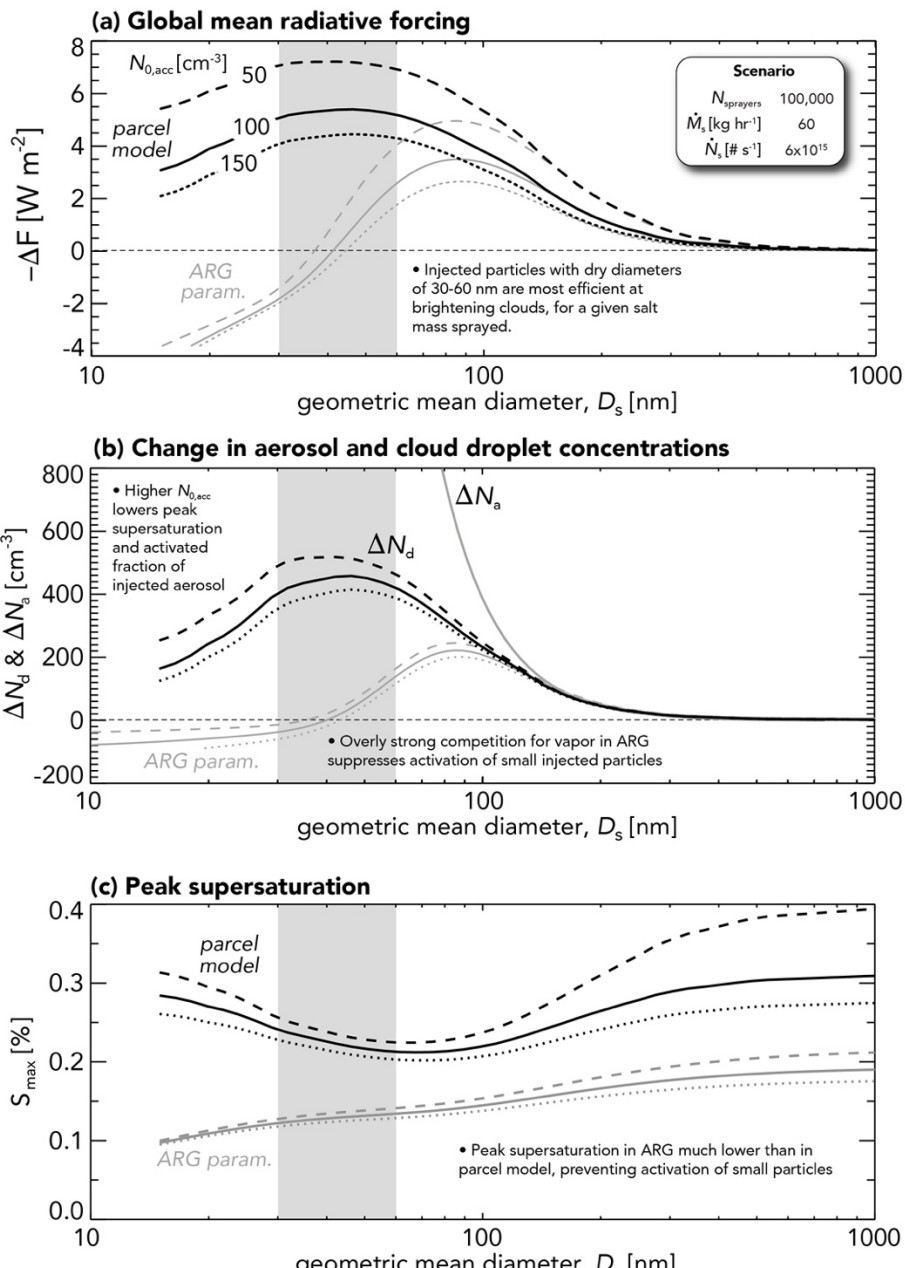

**Figure 7: (a) Global mean radiative forcing** $-\Delta F$ **for a fixed salt mass spray rate (based on Scenario 2, see legend) as a function of injected particle geometric mean diameter** $D_s$ **for three unperturbed accumulation mode aerosol concentrations** $N_{0,acc}$**=50, 100 and 150 cm⁻³. Black curves show the results from the parcel model and gray curves from the ARG parameterization; (b) Change in mean cloud droplet concentration** $\Delta N_d$ **and aerosol concentration** $\Delta N_a$ **in regions where sprayers are operating; (c) Peak supersaturation in the updraft. The gray shaded box indicates the most effective range of** $D_s$**.**






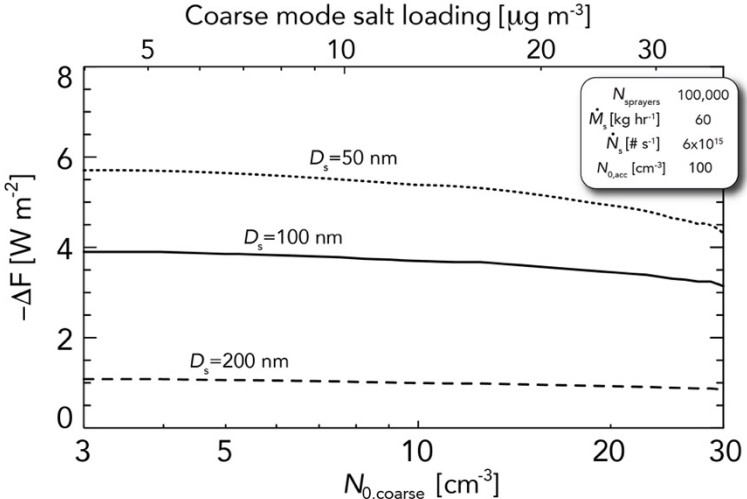

**Figure 8: Global mean radiative forcing $-\Delta F$ for a fixed salt mass spray rate (based on Scenario 2, see legend) as a function of background coarse mode aerosol concentration $N_{0,coarse}$ for three injected particle sizes $D_s$=50, 100 and 200 nm. The equivalent salt mass loading in the coarse mode is indicated by the top axis.**

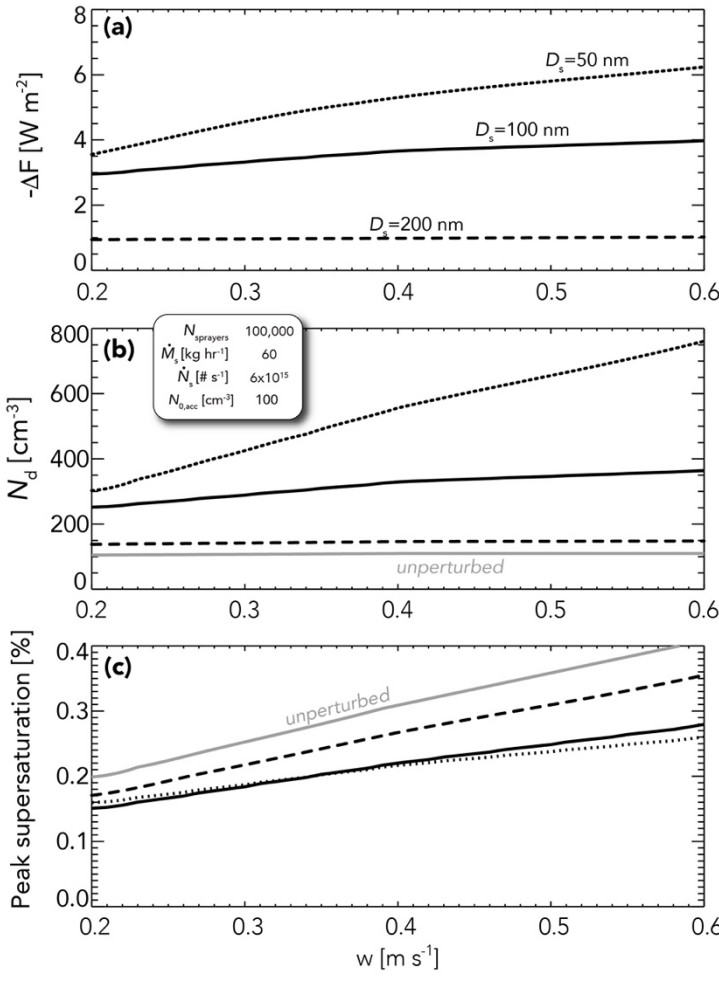

**Figure 9: Impact of assumed updraft speed on (a) global mean radiative forcing $-\Delta F$; (b) cloud droplet concentration $N_d$; (c) peak supersaturation during activation, for a fixed salt mass spray rate (based on Scenario 2, see legend). Results for injected particle sizes $D_s$=50, 100 and 200 nm are shown. Panels (b) and (c) also show the values for the unperturbed case.**







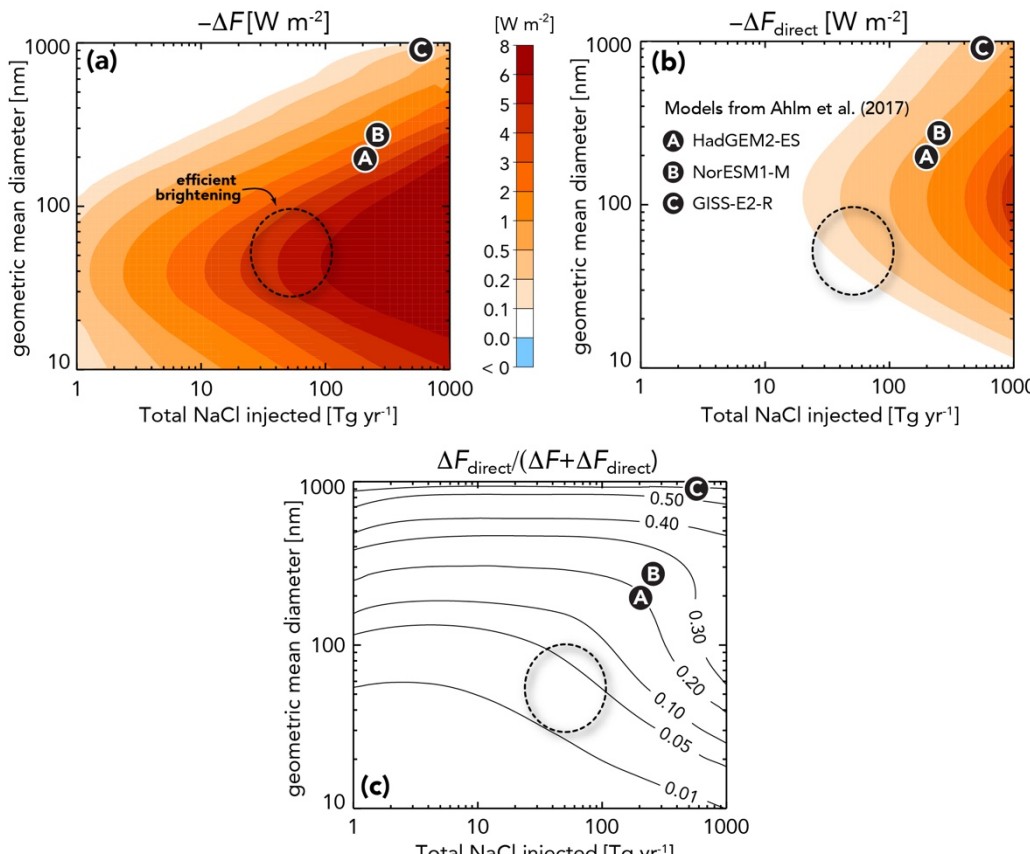

**Figure 10: Global mean radiative forcing from (a) aerosol-cloud interactions (i.e., marine cloud brightening) ΔF; (b) direct radiative forcing of injected aerosol, $\Delta F_{direct}$; (c) ratio of direct to total radiative forcing, plotted as a function of the total salt injection rate and the geometric mean diameter $D_s$ of the injected aerosol. The number of sprayers $N_{sprayers}$=100000, and all other parameters are the same as scenario 2 (see Fig. 5 and Sect. 4.1). The three models used in Ahlm et al. (2017) are shown (see legend in panel b), although it should be noted that for these models injections were confined to the Tropical belt (30ºS-30ºN).**

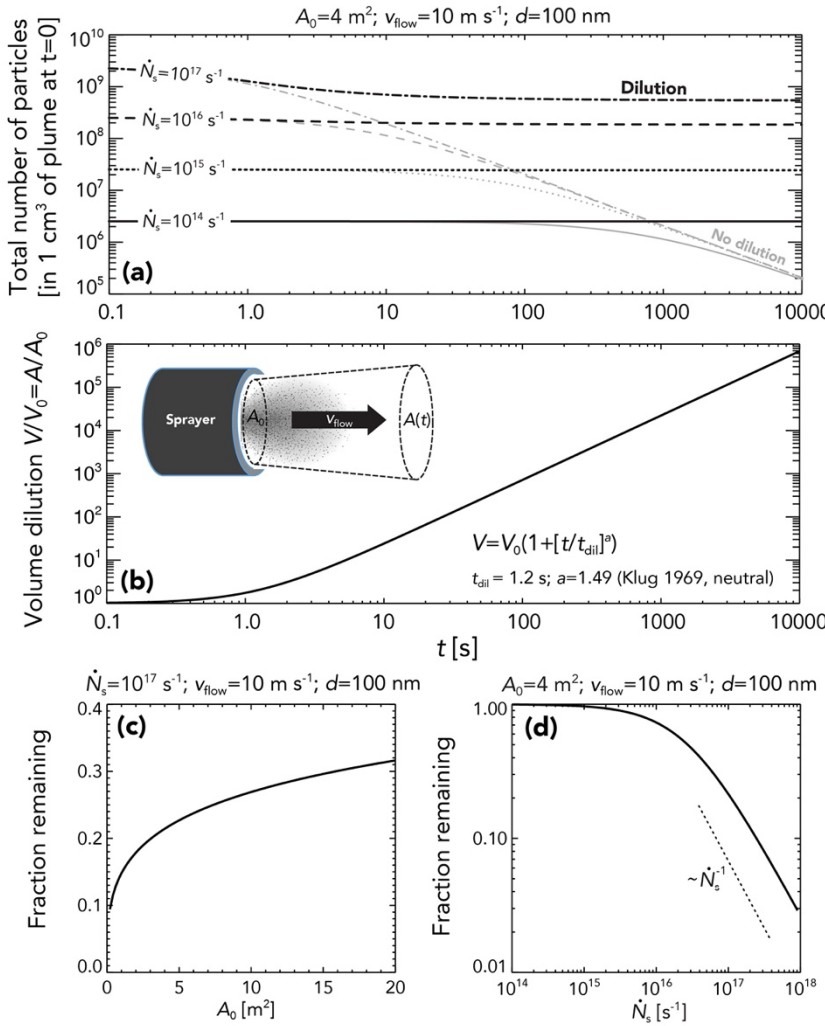

**Figure 11: Effects of coagulation on the concentration of particles at time $t$ downstream of a hypothetical sprayer system with cross sectional area $A=4$ m$^{-2}$ and flow rate of air across the sprayer $v_{flow}=10$ m s$^{-1}$. Here, a single particle diameter $d=100$ nm is assumed, and solutions follow Turco and Yu (1997). (a) Particle concentrations with dilution proceeding according to the dispersion rates for neutral conditions from Klug (1969), as reproduced in Seinfeld and Pandis (2006, Table 18.3); concentrations in the absence of plume dilution are shown for comparison (gray) indicating major losses and eventual asymptotic solution; (b) the ratio of volume $V$ to**
**initial volume $V_0$ increases super-linearly with time with dilution timescale $t_{dil}=1.2$ s; (c) fraction of particles remaining at $t=10,000$ s as a function of initial plume cross sectional area $A_0$ for spray rate $\dot{N}_s=10^{17}$ s$^{-1}$; (d) fraction of particles remaining at $t=10,000$ s as a function of $\dot{N}_s$ for $A_0=4$ m$^2$. The dotted line shows the scaling such that the fraction remaining decreases at the same rate as $\dot{N}_s$ increases, i.e., there is no increase in far-field particle concentration with increasing sprayer output.**


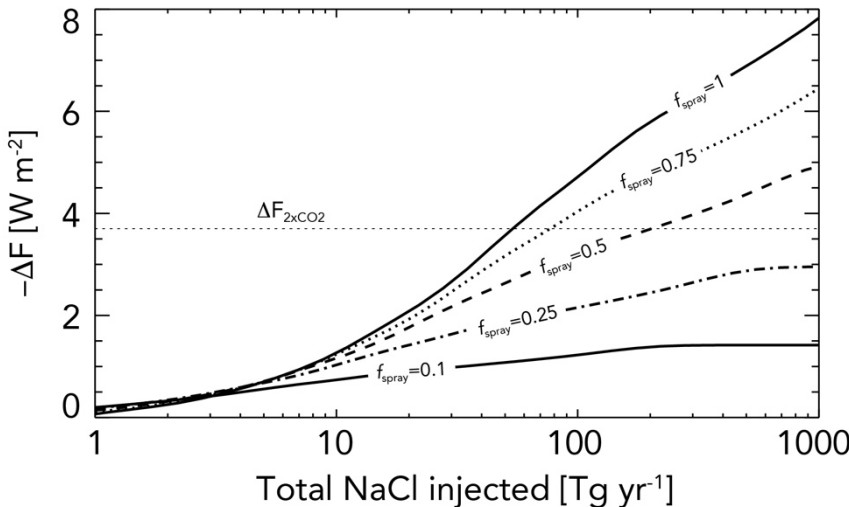

**Figure 12: Sensitivity of the radiative forcing to different fractions $f_{spray}$ of the eligible ocean where sprayers operate. Forcing is shown as a function of the total (global) rate of salt mass injection. The mass injected per sprayer is the same as that for scenario 2 (Sect. 4.1 and Fig. 5) and the sprayer density is the same in each case, i.e., the number of sprayers is proportional to $f_{spray}$ and is 100,000 for $f_{spray}$=1. The geometric mean spray diameters $D_s$=100 nm and the background $N_{0,acc}$ =100 cm$^{-3}$. All other parameters are the same as those used in Fig. 5.**




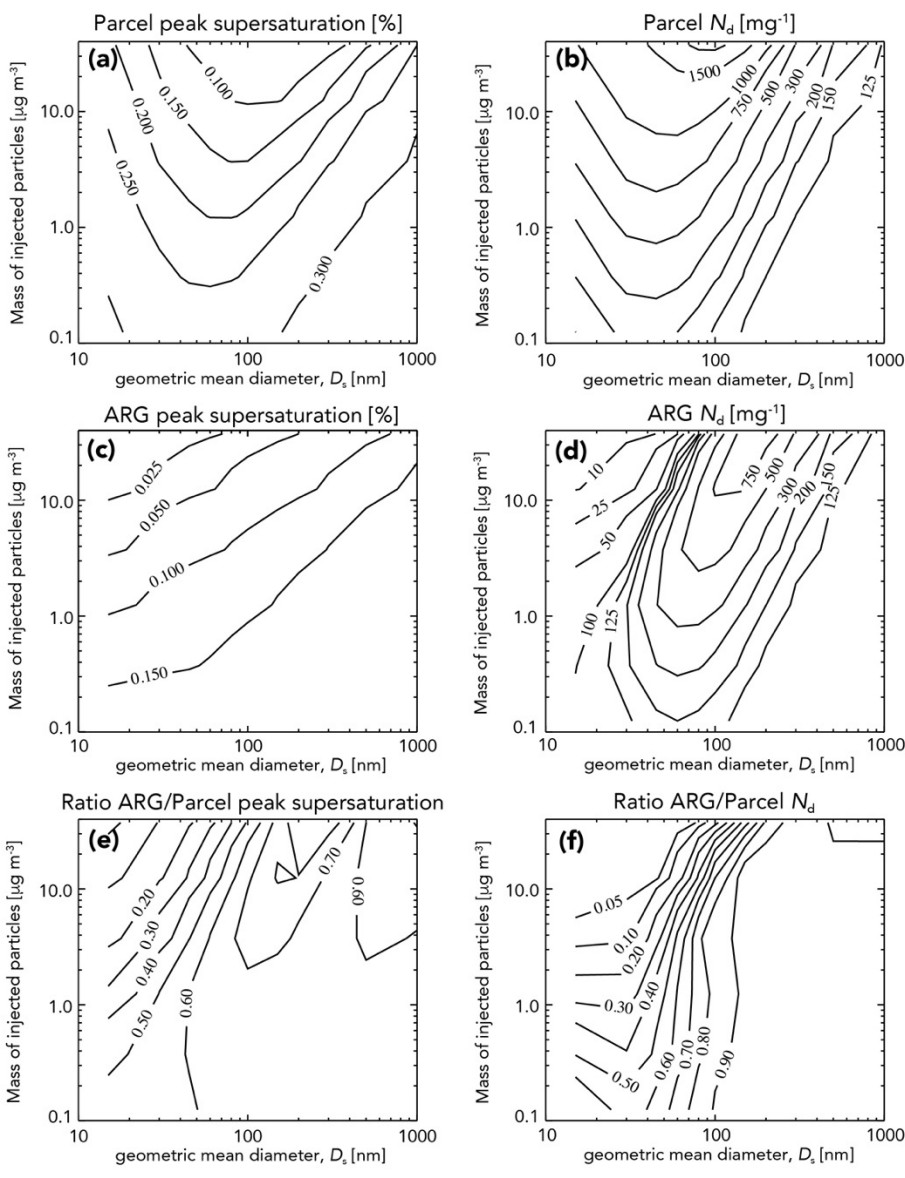


**Figure A1: Comparison of the parcel model and ARG peak supersaturations (panels a, c, e) and activated cloud droplet concentrations $N_d$ (panels b, d, f) as a function of the injected particle geometric mean diameter $D_s$ and the mass loading of injected particles. The other parameters are set as $w= 0.4$ m s$^{-1}$, $N_{0,acc}$=100 mg$^{-1}$, $N_{0,coarse}$= 10 mg$^{-1}$ (base values used throughout the paper). The top row shows results from the parcel model, the center from ARG, and the bottom row is the ratio of ARG to the Parcel model.**
