# Peer review of "Assessing the potential efficacy of marine cloud brightening for cooling Earth using a simple heuristic model"

_Atmospheric Chemistry and Physics, 2021_

## Referee Comment (RC2)

**Review of "Assessing the potential efficacy of marine cloud brightening for cooling Earth using a simple heuristic model" by Wood (acp-2021-327)**

The manuscript investigates the potential of marine cloud brightening (MCB), a geoengineering approach to mitigate global warming by the artificial seeding of clouds with sea salt aerosols to increase their albedo. By developing and applying a heuristic MCB model, the author is able to constrain the effect of various important MCB parameters (the total mass of injected aerosol, the size of the injected aerosol particles, and their number). After that, the author uses these results to discuss implications for other fields currently involved in assessing and developing potential MCB projects (engineering, climate modeling, and large-eddy simulations).

Considering the increasing interest in MCB, not only in science but also in governmental and non-governmental organizations, this manuscript is highly relevant. Moreover, it is very interesting, well-written, and I have only very minor comments, which the author may want to consider. Accordingly, I do not need to see the manuscript again and fully support its publication in *Atmospheric Chemistry and Physics*.

**Minor Comments**

Ll. 47 – 49: A recent paper by Glassmeier et al. (2021) vividly illustrates that the sign of LWP adjustments depends not only on the meteorological conditions but also on the number of aerosol particles, causing positive adjustments when the aerosol number is small and negative adjustments when the aerosol number is large. While I agree that the magnitude of these negative adjustments is probably meteorology dependent, the general trend caused by the number of aerosol particles is probably very relevant to the efficiency of MCB. Therefore, I suggest a short discussion of this effect.

Ll. 49 – 51: Shortly before Ackerman et al. (2004), Wang et al. (2003) also discussed negative LWP adjustments.

Ll. 63 – 65, 620 – 630: In all discussed MCB spraying apparatus, seawater droplets and not aerosol particles are injected into the atmosphere. Since the seawater droplets are slightly larger than aerosol particles, Brownian coagulation might be slower than estimated here.

Ll. 177 – 187: While I believe that I understand what the author is doing here, the last sentence confuses me. I assume that Eq. (4) results from an optimization problem based on MODIS data but not fitting satellite observations. Please clarify.

Ll. 279 – 301: Choosing a Poisson distribution to parameterize track overlap is probably the right choice. However, as long as the spraying vessels do not move, I would assume that tracks are parallel since they are all inside the same boundary layer. Of course, these tracks might overlap if a spraying vessel is directly leeward of a second.

L. 293: Maybe one should note that $n$ is integer.

Ll. 473 – 474: I assume the generally larger size of natural sea spray particles is responsible for their shorter lifetime compared to injected sea salt particles. One should state this clearly.

**Technical Comments**

L. 166: I assume that this should be $\phi_{atm}$ and not $f_{atm}$.

**References**

Ackerman, A. S., Kirkpatrick, M. P., Stevens, D. E., & Toon, O. B. (2004). The impact of humidity above stratiform clouds on indirect aerosol climate forcing. *Nature*, *432*(7020), 1014-1017.

Glassmeier, F., Hoffmann, F., Johnson, J. S., Yamaguchi, T., Carslaw, K. S., & Feingold, G. (2021). Aerosol-cloud-climate cooling overestimated by ship-track data. *Science*, *371*(6528), 485-489.

Wang, S., Wang, Q., & Feingold, G. (2003). Turbulence, condensation, and liquid water transport in numerically simulated nonprecipitating stratocumulus clouds. *Journal of Atmospheric Sciences*, *60*(2), 262-278.

---

## Author Comment (AC1)

**Responses to Reviewer 1 for *Assessing the potential efficacy of marine cloud brightening for cooling Earth using a simple heuristic model* by Robert Wood.**

Reviewer comments in black; responses in red

The manuscript investigates the potential of marine cloud brightening (MCB), a geoengineering approach to mitigate global warming by the artificial seeding of clouds with sea salt aerosols to increase their albedo. By developing and applying a heuristic MCB model, the author is able to constrain the effect of various important MCB parameters (the total mass of injected aerosol, the size of the injected aerosol particles, and their number). After that, the author uses these results to discuss implications for other fields currently involved in assessing and developing potential MCB projects (engineering, climate modeling, and large-eddy simulations).

Considering the increasing interest in MCB, not only in science but also in governmental and nongovernmental organizations, this manuscript is highly relevant. Moreover, it is very interesting, well written, and I have only very minor comments, which the author may want to consider. Accordingly, I do not need to see the manuscript again and fully support its publication in Atmospheric Chemistry and Physics.

I thank the reviewer for their time spent reading and reviewing. I provide detailed responses below.

Minor Comments

Ll. 47 – 49: A recent paper by Glassmeier et al. (2021) vividly illustrates that the sign of LWP adjustments depends not only on the meteorological conditions but also on the number of aerosol particles, causing positive adjustments when the aerosol number is small and negative adjustments when the aerosol number is large. While I agree that the magnitude of these negative adjustments is probably meteorology dependent, the general trend caused by the number of aerosol particles is probably very relevant to the efficiency of MCB. Therefore, I suggest a short discussion of this effect.

Additional sentences are added to the introduction to discuss this result.

Ll. 49 – 51: Shortly before Ackerman et al. (2004), Wang et al. (2003) also discussed negative LWP adjustments.

Excellent point. A citation to Wang et al. (2003) has been added in this location.

Ll. 63 – 65, 620 – 630: In all discussed MCB spraying apparatus, seawater droplets and not aerosol particles are injected into the atmosphere. Since the seawater droplets are slightly larger than aerosol particles, Brownian coagulation might be slower than estimated here.

Adjusted text slightly. The Brownian coagulation kernel is not strongly dependent upon particle size over the range 75-150 nm (it drops by 30% over this range), but the point is well-taken, and a sentence has been added to the manuscript. The particle equilibration time in the sprayer is uncertain.

Ll. 177 – 187: While I believe that I understand what the author is doing here, the last sentence confuses me. I assume that Eq. (4) results from an optimization problem based on MODIS data but not fitting satellite observations. Please clarify.

The idea is to adjust the cloud cover used in the heuristic model to account for the fact that if only a small region is sprayed, one would choose this to be in a region with the highest climatological cloud cover (i.e. subtropical Sc decks). If a greater region is sprayed, the choicest regions have already been taken, so less optimal regions with lower amounts of cloud have to be sprayed. By stacking up the 10x10 degree boxes according to their monthly mean cloud amounts, we can determine the mean low cloud cover as a function of the fraction of ocean sprayed.

Mathematically, if the pdf of low cloud cover is p(f), and the sprayed area fraction ($f_{\text{spray}}$) includes only boxes with cloud cover in excess of some value $f_*$, then one can write:

$$f_{\text{spray}} = \int_{f_*}^{1} p(f)\, df$$

The mean low cloud in these regions is $f_{\text{low}}$ given by:

$$f_{\text{low}} = \int_{f_*}^{1} f p(f)\, df$$

Thus, $f_{\text{low}}$ can be related to $f_{\text{spray}}$ via the parameter $f_*$. Eqn. 4 is determined as a fit to the MODIS data presented in this parametric fashion.

Ll. 279 – 301: Choosing a Poisson distribution to parameterize track overlap is probably the right choice. However, as long as the spraying vessels do not move, I would assume that tracks are parallel since they are all inside the same boundary layer. Of course, these tracks might overlap if a spraying vessel is directly leeward of a second.

The Poisson distribution seems to work similarly even if the tracks are all aligned and overlaps come from sprayers that are too close laterally or are leeward of each other. I was somewhat surprised by this result. I have not yet found a way to use a Poisson type approach if the tracks themselves are not uniform (which we know they are clearly not given that they spread from a point source).

L. 293: Maybe one should note that n is integer.

Done

Ll. 473 – 474: I assume the generally larger size of natural sea spray particles is responsible for their shorter lifetime compared to injected sea salt particles. One should state this clearly.

This is now stated.

Technical Comments

L. 166: I assume that this should be $\phi$!"# and not $f$!"#.

Correct. Thanks. This has been changed in the revised manuscript.

References

Ackerman, A. S., Kirkpatrick, M. P., Stevens, D. E., & Toon, O. B. (2004). The impact of humidity above stratiform clouds on indirect aerosol climate forcing. Nature, 432(7020), 1014-1017.

Glassmeier, F., Hoffmann, F., Johnson, J. S., Yamaguchi, T., Carslaw, K. S., & Feingold, G. (2021). Aerosol-cloud-climate cooling overestimated by ship-track data. Science, 371(6528), 485-489.

Wang, S., Wang, Q., & Feingold, G. (2003). Turbulence, condensation, and liquid water transport in numerically simulated nonprecipitating stratocumulus clouds. Journal of Atmospheric Sciences, 60(2), 262-278.

---

## Author Comment (AC2)

**Responses to Reviewer 1 for** *Assessing the potential efficacy of marine cloud brightening for cooling Earth using a simple heuristic model* **by Robert Wood.**

Reviewer comments in black; responses in red

**General Comments**

This is a good paper on a very current topic, climate engineering. It presents a model to evaluate the multiple different ways in which marine cloud brightening could theoretically be deployed and also suggests ways for improving the simulation of cloud brightening in models. It is well presented and clearly written and I recommend publication once the following comments are addressed.

I thank the reviewer for their review, which has helped improve the manuscript. I provide specific responses to the review comments below.

The paper is generally clear about which physical processes it includes (*e.g.* a detailed treatment of aerosol activation) and which it does not (*e.g.* adjustments of cloud LWP or amount), but one process which is not mentioned and could be of importance is the water injected by the sprayers. The proposals of Latham, Salter *&c.* are for the sprayers to inject a spray of sea-water, not of dry aerosol. Most of the water is assumed to evaporate as the spray is mixed through the depth of the MBL into the cloud above, which with a large number of sprayers has the potential to both moisten and cool the MBL with subsequent impacts on the cloud layer. A discussion of this process should be included and the impact of omitting it assessed.

The reviewer raises an important topic that will have significant implications for the design of particle spray systems needed to implement any marine cloud brightening (MCB). Seawater contains only 3.5% sodium chloride by mass, but the equilibrium mass concentration of salt in solution droplets close to the surface in the marine boundary layer is closer to 30% (given a typical relative humidity of ~80%). For the small sizes of droplets proposed (~100 nm), the timescale for reaching equilibrium size is on the order of seconds or less, and so a considerable amount of water vapor will be evaporated from the seawater droplets close to the spray system. This has the potential to lead to an aerosol with negative buoyancy that is unable to rise to cloud base and interact with clouds.

Although not currently stated in the submitted manuscript, it is assumed that this problem can be overcome either by rapid dilution of the particle laden air immediately downwind of the spray system, and potentially by the addition of thermal energy sufficient to overcome the evaporative cooling and maintain neutral or slightly positive

buoyancy. I see this as essentially an engineering challenge that is beyond the scope of this paper. A statement is now included to acknowledge the

The concern that the injections of additional water may cool and moisten the PBL sufficiently to influence the clouds, we can consider the energy and moisture budgets of the PBL. In the manuscript, we show that salt mass injection rates for a single sprayer may be as high as 660 kg of salt per hour (Fig. 5 and associated discussion). Given the 3.5% sodium chloride concentration in seawater, this is a water injection rate of ~18,000 kg per hour or 5 kg $s^{-1}$. We can compare this with the natural source of evaporated water from the surface (evaporated flux) if we assume an area over which the additional sprayed water influences the PBL. It is reasonable to choose the assumed track area ($5.28 \times 10^{10}$ $m^2$, see section 2.5 in the manuscript) as the area being influenced by a single sprayer. Given a typical surface latent heat flux of 100 W $m^{-2}$, or equivalently an evaporated water flux of $4 \times 10^{-5}$ kg $m^{-2}$ $s^{-1}$, the additional injected water contributes $5/5.28 \times 10^{10} \sim 10^{-10}$ kg $m^{-2}$ $s^{-1}$, or $1/500,000^{th}$ of the natural evaporated water flux from the surface. This is essentially a negligible contribution of the injected water to the PBL moisture budget. A similar argument can be made for the sprayer contribution to the PBL energy budget. Similarly negligible contributions of heat and moisture injections from container ships were shown to be the case based on the Monterey Area Ship Track Experiment (MAST) in 1994 (see Hobbs et al., 2000). The impact of injected moisture and resultant cooling is thus essentially an issue affecting the very near field buoyancy (within ~100 m of the spray system) and will have negligible impact on PBL moisture and temperature budgets as a whole.

A statement is now included in the revised manuscript to clarify this.

Hobbs, P. V., Garrett, T. J., Ferek, R. J., Strader, S. R., Hegg, D. A., Frick, G. M., Hoppel, W. A., Gasparovic, R. F., Russell, L. M., Johnson, D. W., & others. (2000). Emissions from ships with respect to their effects on clouds. Journal of the Atmospheric Sciences, 57(16), 2570–2590.

**Specific Comments**

**Lines 65-69 and 75-76.** The forcing of -0.06 to -0.6 Wm$^{-2}$ due to the current commercial fleet of around 60,000 vessels stands in sharp contrast to the forcing of around -4 Wm$^{-2}$ due to only ~10,000 MCB sprayers (these latter results are introduced later in the manuscript). It would be nice if, somewhere later in the paper, this apparent contradiction could be discussed (presumably the issue is particle size).

This apparent contradiction is interesting and worthy of further investigation. Total $SO_2$ emissions from shipping are ~10 Tg $yr^{-1}$ (see Lauer et al., 2007). Assuming this is all converted into sulfate, this equates to a mass of 15 Tg $SO_4$ $yr^{-1}$. From Figure 5 in the

manuscript, introducing injections of 15 Tg of NaCl per year would yield a Twomey radiative forcing of ~1.5-2.0 W m$^{-2}$ assuming the particles have a modal dry diameter of 100 nm. This represents a considerably higher efficacy (per mass of solute) for MCB compared with commercial shipping. Although sea salt is a more effective CCN than sulfate, both species are highly hygroscopic, and so I do not believe that this argument can explain the greater efficacy. However, observations show that accumulation mode particles over the oceans, which consist mostly of sulfate, tend to be closer to 200 nm diameter than to 100 nm diameter (e.g., Heintzenberg et al., 2000). Although commercial ships do emit a considerable number of small particles, over the lifetime of the emitted SO2, one would expect considerable growth into the accumulation mode. One hypothesis to explain the greater efficacy of MCB is that commercial shipping emissions result in larger accumulation mode particles. Fig. 6 in the manuscript shows that 15 Tg yr$^{-1}$ of injected salt particles with a modal diameter of 200 nm would yield a radiative forcing of only 0.5 W m$^{-2}$, a value closer to the estimates for marine shipping. In addition, shipping emissions are much more concentrated geographically than those from our heuristic model (for $f_{spray}$=1), which assumes an essentially random distribution of sprayers distributed over the eligible regions (remote regions) of the global oceans. A complete treatment of the effects of geographical heterogeneity of the shipping sulfate is beyond this response but may also contribute significantly.

The revised manuscript now includes a brief discussion of this.

Lauer, A., Eyring, V., Hendricks, J., Jöckel, P., & Lohmann, U. (2007). Global model simulations of the impact of ocean-going ships on aerosols, clouds, and the radiation budget. *Atmospheric Chemistry and Physics*, *7*(19), 5061–5079.

Heintzenberg, J., Covert, D. C., & Van Dingenen, R.: Size distribution and chemical composition of marine aerosols: A compilation and review. Tellus B, 52(4), 1104–1122, 2000.

**Lines 110 vs. 118**. The former says Rasch *et al*. used an $N_d$ value of 1000 cm$^{-3}$ but the latter says they increased $N_d$ to 375 cm$^{-3}$.

Thank you for spotting this. The second mention should have stated 1000 cm$^{-3}$. This has now been corrected

**Lines 181-182**. It should be made clear that *f_spray* is an input parameter to the model, not something which has been determined from observations. It should also be emphasised that *f_spray* and *f_low* are single (global) parameters because their description follows immediately after a discussion which talks about 10x10 degree gridboxes and it's easy for the reader to continue with that idea and imagine a

geographic distribution of values of *f_spray* and *f_low* with different values in each gridbox.

Good point. The revised text has been edited to make this clearer.

**Line 191-192**. This needs a little rewriting to avoid the impression that the second sentence (beginning "Cloud condensate...") is still referring to the areas without low cloud which are referred to in the first sentence.

Rewritten to improve clarity.

**Lines 205-206**. Does the phrase "Assuming the entire ocean area could be seeded" imply an *f_spray* value >1? Otherwise I can't see how an $r_N$ value of 2.4 can give a forcing of -3.7 Wm$^{-2}$ from Fig.1. Please make this clear if it is indeed the case (or if not, explain where the $r_N$ value of 2.4 comes from).

This would be the case if $f_{ocean} = 0.7$, $f_{spray} = 1$, which is now clarified in the manuscript. We do this to facilitate comparison with the Slingo study.

**Line 218**. How sensitive are the results to the assumption of stationary sprayers? Most promotional material I've seen for such sprayers depicts them as decidedly non-stationary ships.

This is difficult to assess. If sprayers move in the same direction and with the same speed as the wind, then the aerosol would be highly concentrated in the vicinity of the ship. The Twomey effect would be very strong near the ship, but it would not be felt over a large area. In general, this would reduce the efficacy of the forcing. On the other hand, if sprayers move in the opposite direction to the wind, the particles would be more rapidly distributed and the overall effectiveness would increase. Given that it is impossible for the sprayers to *always* be opposing the wind direction (which tends to be quite steady in the regions with the greatest low cloud amounts), it seems unlikely that the global MCB forcing will be influenced to first order by sprayer motion. That said, sprayer placement and motion for effective implementation of MCB would need to be optimized based on our knowledge of cloud regime susceptibility, climatology, background aerosol, and other meteorological factors. Spray rates and sprayer motion could be tweaked in real time based on forecast models.

**Lines 264-265 and Fig.2(c)**. Is the value of 0.41 TJ for the "Spreading" line the amount for the 0-3 day period shown in the figure or the amount estimated for the whole duration of the perturbation (perhaps as long as a week, given the form of the curve shown in the figure)?

The 0.41 TJ is for the whole duration of the perturbation. The plot only shows a limited portion, but the model is run out to 10 days to obtain this value. This is now clarified in the caption of Fig. 2.

**Line 315 (and elsewhere)**. "PBL" is introduced (without definition) in the context "marine PBL" where it seems "MBL" (which has been defined) would do just fine. "PBL" also turns up in lines 348, 364, 438 and both tables. Either use "MBL" throughout or define PBL.

We have used MBL uniformly through the revised manuscript.

**Line 321**. As the source of the injected aerosols is presumably the local seawater it is unlikely to be pure sodium chloride.

Although seawater contains other species, for the inorganic components that dominate the mass of bulk seawater, a recent study suggests that a hygroscopicity parameter only slightly smaller than that of pure sodium chloride is appropriate (kappa=1.1 vs 1.2). Tests show that the results are rather insensitive to changing kappa from 1.2 to 1.1. A statement has been included in the manuscript.

Zieger, P., Väisänen, O., Corbin, J. C., Partridge, D. G., Bastelberger, S., Mousavi-Fard, M., Rosati, B., Gysel, M., Krieger, U. K., Leck, C., Nenes, A., Riipinen, I., Virtanen, A., & Salter, M. E. (2017). Revising the hygroscopicity of inorganic sea salt particles. *Nature Communications*, *8*(1), 15883. https://doi.org/10.1038/ncomms15883

**Line 336 and Eq.(11)**. The greek letter tau has already been used (with the suffix "res") to denote a timescale, whereas here (with a different suffix) it's used for AOD. I suggest using different symbols for two such very different quantities.

AOD is now used instead of the Greek tau for aerosol optical depth.

**Lines 366-367**. The phrase "This is handled in the heuristic model as described in Sect. 2.5" really isn't sufficient: Section 2.5 has a lot in it so the process used needs to be spelled out in a bit more detail.

This is an excellent point. An additional paragraph is included to provide more details on how the LES tracks are represented in the heuristic model.

**Lines 473-4**. The reason for the "considerably shorter" residence time of natural sea spray particles should be given (larger particle sizes?).

Done

**Lines 596-7**. For clarity, this sentence needs to end something like this: "....required to produce a significant radiative forcing via cloud modificaton rather than direct aerosol forcing." The point being that "significant radiative forcing" can in principle be produced via either mechanism depending on the size ($D_s$) of the particles injected.

Good point. Corrected now.

**Line 661.** "suggests" would be more appropriate than "demonstrates" - all Fig.12 actually demonstrates is that the minimum value of $f\_spray$ required to achieve a -dF of 4 Wm$^{-2}$ is somewhere between 0.25 and 0.5.

Good point. Changed to "suggests"

**Line 689**. "insufficiently short" means "too long" which is precisely the opposite of what is intended.

Good spot. Changed to "...are of insufficient duration..."

**Line 713**: It would aid clarity if this was written out in full: "...tend to produce more brightening for clean cases and less for polluted cases".

Done

**Technical Corrections**

Thank you for the following, which have been corrected in the revised manuscript.

**Line  53**: "handled" not "handed".

**Line 166**: "$\varphi_{atm}$" not "$f_{atm}$".

**Line 264**: Remove the underlining of "8" in "(Eq.8)".

**Line 436**: An extra ")" is required after "(5)" to close the parenthesis opened on the previous line.

**Line 498**: "...of very adding small..." should be "...of adding very small...".

**Line 705**: "presents" not "presented".

**Line 739**: Shouldn't this line have a bullet point?

**Figure 5(a)**: Why include the blue (negative) segment on the color-bar? It's not needed.